METHODS AND RESOURCES

# The Brain/MINDS Marmoset Connectivity Resource: An open-access platform for cellular-level tracing and tractography in the primate brain

Henrik Skibbe[1]*, Muhammad Febrian Rachmadi[1], Ken Nakae[2], Carlos Enrique Gutierrez[3], Junichi Hata[4,5], Hiromichi Tsukada[3,6], Charissa Poon[1], Matthias Schlachter[1], Kenji Doya[3], Piotr Majka[7,8,9], Marcello G. P. Rosa[8,9], Hideyuki Okano[4,5], Tetsuo Yamamori[10,11], Shin Ishii[12], Marco Reisert[1,13,14,15], Akiya Watakabe[10]

1 Brain Image Analysis Unit, RIKEN Center for Brain Science, Wako, Saitama, Japan, 2 Exploratory Research Center on Life and Living Systems (ExCELLS), National Institutes of Natural Sciences, Aichi, Japan, 3 Neural Computation Unit, Okinawa Institute of Science and Technology Graduate University, Onna Village, Japan, 4 Laboratory for Marmoset Neural Architecture, RIKEN Center for Brain Science, Wako, Saitama, Japan, 5 Department of Physiology, Keio University School of Medicine, Tokyo, Japan, 6 Center for Mathematical Science and Artificial Intelligence, Chubu University, Kasugai, Aichi, Japan, 7 Laboratory of Neuroinformatics, Nencki Institute of Experimental Biology of Polish Academy of Sciences, Warsaw, Poland, 8 Australian Research Council, Centre of Excellence for Integrative Brain Function, Monash University Node, Clayton, Australia, 9 Neuroscience Program, Biomedicine Discovery Institute and Department of Physiology, Monash University, Clayton, Australia, 10 Laboratory of Haptic Perception and Cognitive Physiology, RIKEN Center for Brain Science, Wako, Saitama, Japan, 11 Department of Marmoset Biology and Medicine, Central Institute for Experimental Animals, Kawasaki, Japan, 12 Department of Systems Science, Kyoto University, Kyoto, Japan, 13 Department of Stereotactic and Functional Neurosurgery, Medical Center of the University of Freiburg, Freiburg Im Breisgau, Germany, 14 Medical Faculty of the University of Freiburg, Freiburg Im Breisgau, Germany, 15 Department of Diagnostic and Interventional Radiology, Medical Physics, Medical Center–University of Freiburg, Freiburg Im Breisgau, Germany

* henrik.skibbe@riken.jp

**Data Availability Statement:** Links for accessing the BMCR-Explorer (http://bmca.riken.jp/), as well as for downloading the NIfTI files, data and tools,

# Abstract

The primate brain has unique anatomical characteristics, which translate into advanced cognitive, sensory, and motor abilities. Thus, it is important that we gain insight on its structure to provide a solid basis for models that will clarify function. Here, we report on the implementation and features of the Brain/MINDS Marmoset Connectivity Resource (BMCR), a new open-access platform that provides access to high-resolution anterograde neuronal tracer data in the marmoset brain, integrated to retrograde tracer and tractography data. Unlike other existing image explorers, the BMCR allows visualization of data from different individuals and modalities in a common reference space. This feature, allied to an unprecedented high resolution, enables analyses of features such as reciprocity, directionality, and spatial segregation of connections. The present release of the BMCR focuses on the prefrontal cortex (PFC), a uniquely developed region of the primate brain that is linked to advanced cognition, including the results of 52 anterograde and 164 retrograde tracer injections in the cortex of the marmoset. Moreover, the inclusion of tractography data from diffusion MRI allows systematic analyses of this noninvasive modality against gold-standard cellular connectivity data, enabling detection of false positives and negatives, which provide a basis for

are provided on the Brain/MINDS data portal at: https://dataportal.brainminds.jp/marmoset-connectivity-atlas. The image data is also openly available as NIfTI files on the RIKEN CBS data portal at: https://doi.org/10.60178/cbs.20230630-001. The source code for both the processing pipeline and figure generation can be accessed publicly at: https://doi.org/10.5281/zenodo.7906530 and https://doi.org/10.5281/zenodo.7906607, respectively.

**Funding:** This work was supported by the program for Brain Mapping by Integrated Neurotechnologies for Disease Studies (Brain/MINDS) from the Japan Agency for Medical Research and Development AMED: grant number JP15dm0207001 to H.S., M.F.R., J.H., C.P., H.O., S.I., T.Y., A.W, grant number JP19dm0207088 to K.N., and grant number JP18dm0207030 to K.D. T.Y. is supported by Scientific Research on Innovative Areas (22123009) from MEXT, Japan. P.M. is supported by the National Science Centre of Poland (2019/35/D/NZ4/03031). The data from the Marmoset Brain Connectivity Atlas were obtained with support from Australian Research Council (DP110101200, DP140101968, CE140100007) and National Health and Medical Research Council (APP1194206) to MGPR. JSPS KAKENHI (grant numbers JP22H05163 and JP22H05154 to K.N., JP20H03630 to J.H., \ JP22H04998 to S.I., and JP22K15658 to C.P.). The funders had no role in study design, data collection and analysis, decision to publish, or preparation of the manuscript.

**Competing interests:** The authors have declared that no competing interests exist.

**Abbreviations:** BMCR, Brain/MINDS Marmoset Connectivity Resource; CD, caudate nucleus; dMRI, diffusion-weighted MRI; HARDI, high angular resolution diffusion imaging; PFA, paraformaldehyde; PFC, prefrontal cortex; STPT, serial two-photon tomography.

future development of tractography. This paper introduces the BMCR image preprocessing pipeline and resources, which include new tools for exploring and reviewing the data.

## Introduction

To better understand the function of the primate brain, it is essential to map its connectivity at the cellular level. Since mapping an entire mammalian brain at single synapse resolution remains impractical due to various technical reasons, the optimal combination of sensitivity and specificity for systematically mapping connectomes is currently achieved through neuronal tracer injections combined with high-resolution fluorescence microscopy.

One of the most extensive open tracer image databases for mammalian brain connectivity is the Allen Mouse Brain Connectivity Atlas [1,2], which is the current standard for collecting, processing, and publicly sharing brain connectivity data from animal models. However, using rodents to understand primate cognition has limitations, which are related to differences in brain anatomy that result in less complex cognitive abilities [3,4].

Mental and neurological disorders, including age-related dementia, pose a major challenge to modern societies, with broad implications for economic development and well-being. Therefore, it is not surprising that there is great interest in studying the structure and function of primate brains to advance our understanding of the origin, development, and treatment of such diseases.

In recent years, marmosets have gained popularity as primate models due to their small size and high reproductive rate, coupled with essential anatomical, physiological, and cognitive characteristics that differentiate primate brains [5–7]. For example, the marmoset has become a model for studying Parkinson's disease [8], autism spectrum disorder [9], and Alzheimer's disease [10]. Unlike rodents, marmosets have well-developed visual and auditory cortices, which contain the same basic subdivisions as the human brain and reflect specializations for social interaction [3,11–13], a complex of premotor and posterior parietal areas responsible for sophisticated spatial and movement planning functions [14,15], and the same basic subdivisions of the prefrontal cortex (PFC) as the human brain [16,17]. A surge of interest in marmosets has led to the development of various neuroinformatic resources, including the Marmoset Brain Mapping project [18,19], the marmoset Brain/MINDS Atlas [20], and the Marmoset Brain Connectivity Atlas [21]. The latter comprises a large amount of cortical retrograde tracer data [21,22].

Here, we introduce the implementation and features of the Brain/MINDS Marmoset Connectivity Resource (BMCR), a public access resource that provides a significant new step towards the exploration of the structural basis of primate cognition. The BMCR was constructed using datasets from the Brain/MINDS project [23], which consist of TET-amplified AAV anterograde neural tracer injections into various locations in the marmoset PFC, a key region that differentiates primates from other mammals. This core database contains data from 52 anterograde neural tracer injections in adult marmosets and has complementary structural MR images for 23 of them. Further, for 19 of the datasets, we combined a retrograde tracer with the anterograde tracer, resulting in the ability to visualize bidirectional connections. This paper describes the post-processing and validation of the data, making it accessible to a broader community of non-imaging experts and provides access to tools such as the BMCR-Explorer and Nora StackApp.

Automated serial two-photon tomography (STPT) [24] was used to acquire serial section images of the fluorescent anterograde tracer signals. Coronal sections were taken every 50 μm,

with an in-plane resolution of about 1.35 μm/px, which is sufficiently high to identify individual axon structures in the imaging plane. In addition, backlit images were taken before Nissl staining from sections that were collected after two-photon tomography to reveal features of the brain myelination. Nissl and backlit sections were imaged under brightfield microscopy. Prior to STPT acquisition, ex vivo whole brain MR images were acquired from marmosets using the high angular resolution diffusion imaging (HARDI) technique. All images were automatically processed, and the results were integrated into the BMCR. The BMCR gives access to the datasets in a common reference image space with a high resolution of $3{\times}3{\times}50~\mu m^3$ that shows detailed morphology of axon fibers (Fig 1). Tools and supplementary data, such as atlas annotations and diffusion-weighted MRI (dMRI) measurements, are also provided in the BMCR.

The present resource includes mappings to and from the image spaces used in the aforementioned resources, demonstrated in the present paper by integration with the retrograde tracer data from the Marmoset Brain Connectivity Atlas. Thus, the BMCR not only provides a new dataset for understanding PFC connectivity, but also a data transfer system for integrating other databases.

## Results

### The BMCR image processing pipeline

A major component of the work that enabled the BMCR is its image post-processing pipeline, the first of its kind for processing STPT images of entire marmoset brains. The pipeline includes fully automated processing of tracer signals, including the detection of the injection site, and the segmentation of anterograde and retrograde tracer signals from the tissue background. It incorporates the mapping of data to a reference image space in high resolution $(3{\times}3{\times}50~\mu m^3)$ and introduces a new cortical flatmap stack mapping. A flatmap stack is a 3D image representation of the marmoset cortex that extends cortical flatmaps with representations of cortical depth. Flatmap stack mappings are extensions of the flatmaps that are part of the Marmoset Brain Mapping atlases, the Brain/MINDS Atlas, and Marmoset Brain Connectivity Atlas. The next sections briefly summarize the data and the pipeline output. Fig 2 shows an overview.

All data were visually screened by an expert before being made publicly available using the BMCR-Explorer and Nora-StackApp visualization tools; both tools are explained in more detail later. In addition, we have performed quantitative validations showing that automatic injection site localization and image registration to the template space are equivalent or even outperform manual mapping. Details about the pipeline and validation can be found in the Methods section.

**Pipeline inputs.** We obtained data from up to 4 kinds of image modalities from single marmoset brains: dMRI for fiber tracking, STPT for anterograde tracing, light microscopy for Nissl and backlit images, and fluorescent microscopy for retrograde tracing. The core data of the BMCR are derived from 52 individuals, 33 female and 19 male.

For each individual, TET-amplified AAV neural tracer injections [25–27] were administered to a single brain region within the left hemisphere of the marmoset PFC. See the Methods section for details. Two kinds of ex vivo full-brain dMRI images were acquired with a 9.4 Tesla MRI animal scanner (Bruker Optik GmbH, Germany). The first image was a T2-weighted (T2W) MRI with a resolution of $0.1{\times}0.1{\times}0.2~mm^3$, while the second image was acquired using the HARDI protocol (b-values of 1,000, 3,000, and 5,000 $s/mm^2$, isotropic resolution of 0.2 mm, and 128 independent diffusion directions). After MRI acquisition, images of 50 μm coronal sections revealing the tracer signal in the entire brain were acquired automatically with

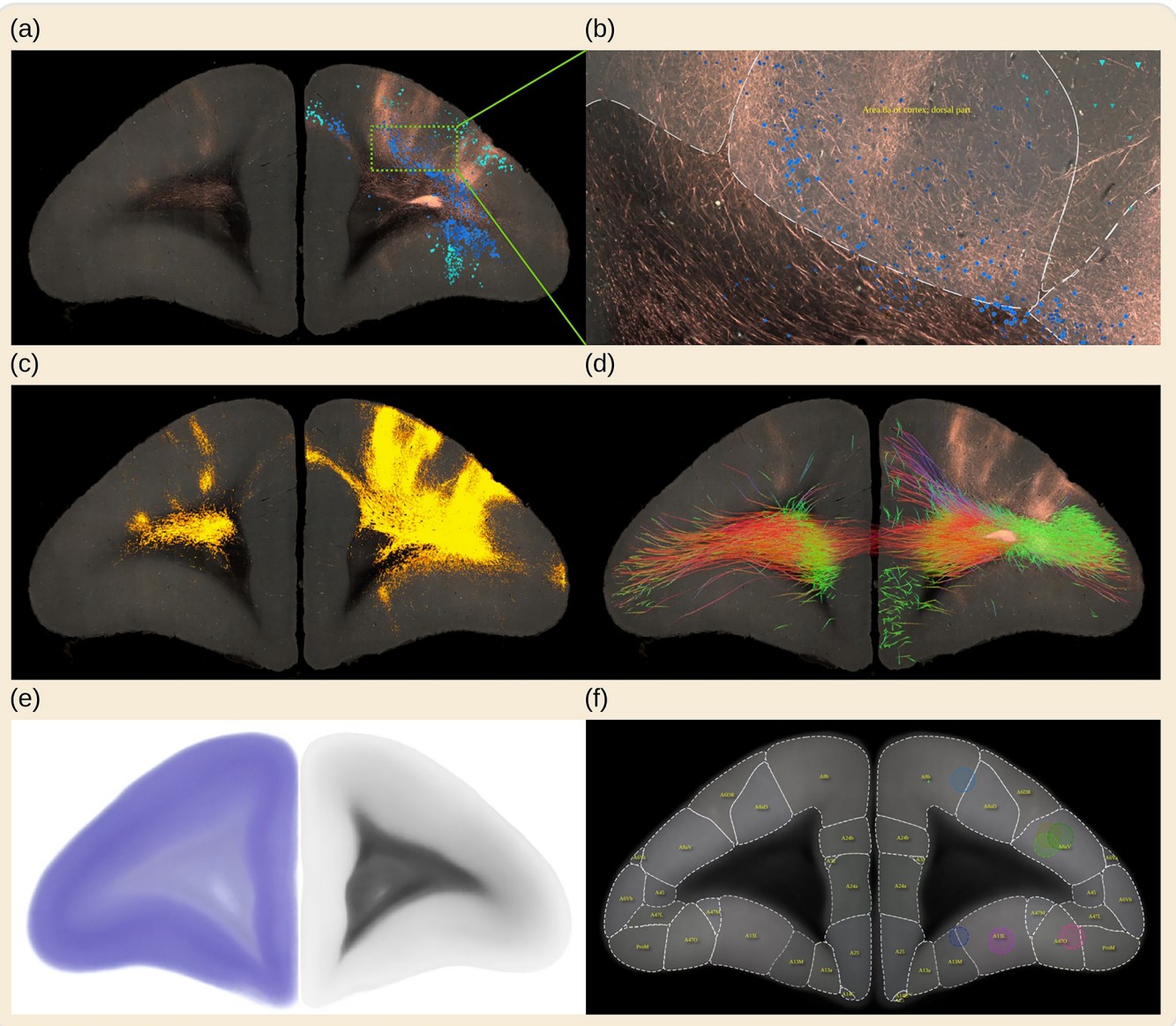

**Fig 1. BMCR example images.** The BMCR comprises the image data of 52 anterograde tracer injections and 19 retrograde tracer injections placed into the marmoset PFC, supplemented by retrograde neural tracer data from the Marmoset Brain Connectivity Atlas project (145 injections). This figure shows a series of examples of virtual coronal sections from the BMCR all in the same reference image space. The tracers were injected into Area 8a of the cortex. Panel (a) shows an STPT fluorescent image of the results of an anterograde neural tracer, and as an overlay, neurons labeled by a retrograde tracer dataset from the Marmoset Brain Connectivity Atlas; this dataset originated from a similarly located injection site, mapped to the BMCR image space (the different blue tones, blue and turquoise) indicate whether cells are beneath or above layer IV, information provided by the BMCR. Panel (b) shows a detailed close-up of a portion of the same image. Panel (c) shows the segmented tracer from (a) over the auto-fluorescent background. The overlay in panel (d) shows tractography results of streamlines originating from the same site as the tracer injection, based on averaged dMRI data. The colors reflect streamline directions. The BMCR also includes individual and population average backlit and Nissl images (see (e) for examples of the population average images) and incorporates brain region annotations from major brain atlases for marmosets (see (f) for an example of the Brain/MINDS Atlas). BMCR, Brain/MINDS Marmoset Connectivity Resource; dMRI, diffusion-weighted MRI; PFC, prefrontal cortex; STPT, serial two-photon tomography.

TissueCyte STPT (TissueVision, Cambridge, Massachusetts, United States of America). The spatial resolution was 1.385 μm×1.339 μm. For 19 of the tracer injections, we added AAV2re-tro-EF1-Cre, a non-fluorescent retrograde neural tracer. After STPT imaging, every 10th section was collected and fluorescently labeled for Cre immunoreactivity. Then, the sections were imaged with an all-in-one microscope (Keyence BZ-X710, Japan). Another set of sections was

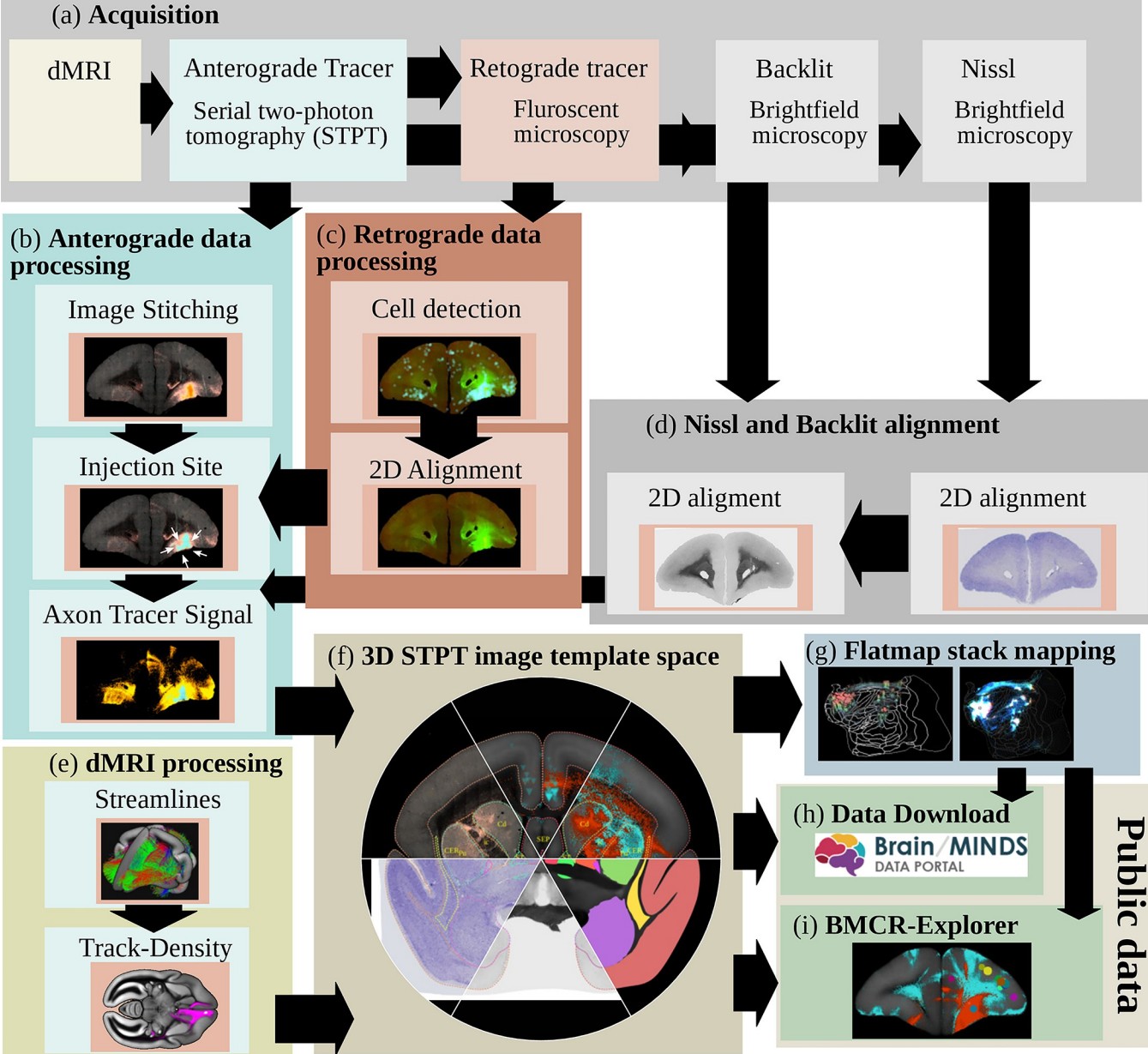

**Fig 2. The BMCR image post-processing pipeline.** The steps of the BMCR image post-processing pipeline: (a) Image acquisition: After dMRI imaging and automated STPT imaging and sectioning using 2p-tomography, retrograde tracer, Nissl and backlit images are taken. (b) Processing and analysis of anterograde imaging data, and (c) retrograde imaging data, respectively. (d) Automated alignment of Nissl and backlit images. (e) Track-density images are generated from streamlines representing major axon fiber bundles touching the injection site. (f) All data, including high-resolution microscopy data are mapped to the BMCR 3D brain reference image space. The final steps (g) are creating the flatmap stack, (h) preparing the data for downloading, and (i) integrating it into the BMCR-Explorer. BMCR, Brain/MINDS Marmoset Connectivity Resource; dMRI, diffusion-weighted MRI; STPT, serial two-photon tomography.

collected and imaged twice: first, as backlit images to reveal features of myelination, and second, after Nissl staining. We used the same microscope for both backlit and Nissl images (Keyence BZ-X710). The pixel resolution of these sections was 3.774 μm/px.

**Pipeline outputs.**   The pipeline's purpose was to detect and segment neural tracer signals in the images, perform fiber tracking, and integrate all data into a common reference space.

**Table 1. Data summary.**

| Resource | Number | Description | Isotropic | Highres |
|---|---|---|---|---|
| Anterograde tracer | 52 | STPT (3 channels) | 50 μm | 3×3×50 μm$^3$ |
| Tracer density | 52 | tracer positive voxels | 50 μm | 3×3×50 μm$^3$ |
| Tracer intensity | 52 | masked channel 2 of STPT image | 50 μm | |
| Injection site cell location | 52 | csv (text) | | |
| Injection site cell density | 52 | density image | 50 μm | |
| Nissl images | 52 | brightfield microscopy | 50 μm | 3×3×50 μm$^3$ |
| Backlit images | 52 | brightfield microscopy | 50 μm | 3×3×50 μm$^3$ |
| Retrograde tracer cell location | 19 | csv (text) | | |
| HARDI (dMRI) | 23 | HARDI protocol, 128 directions | 200 μm | |
| STPT template | 1 | population average | 50 μm | |
| Nissl template | 1 | population average | 50 μm | |
| Backlit template | 1 | population average | 50 μm | |
| HARDI (dMRI) template | 1 | population average, 64 directions | 100 μm | |
| Brain/MINDS atlas | 1 | GM annotations | 50 μm | |
| Marmoset Brain Mapping v2 and v3 atlases | 1 | GM and WM annotations | 50 μm | |
| Marmoset Brain Connectivity Atlas | 1 | cortical GM annotations | 50 μm | |
| | 145 | retrograde tracer data (json) | | |

BMCR, Brain/MINDS Marmoset Connectivity Resource; dMRI, diffusion-weighted MRI; STPT, serial two-photon tomography.

Table 1 lists the available data. The data are complemented by external resources that have been mapped to the same image space.

The image processing pipeline automatically reconstructed a 3D image stack from the microscopy images, identified injection site locations, and employed a deep neural network [28] to localize cell body positions within the injection site and segment tracer signals from the background. It then mapped all data, including Nissl and backlit images, to a population average STPT brain image that we used as a template image space. The average STPT brain image was generated by iteratively registering 36 subjects (including their left/mirrored versions) using the ANTs image registration toolkit [29]. The pipeline mapped all microscopy images to an isotropic 50 μm and 100 μm voxel resolution. In addition, it mapped the data to the STPT template in high resolution. The target resolution for high-resolution data was 3×3×50 μm$^3$ leading to detailed, co-registered full-brain image stacks with a size of 9,666×8,166×800 voxels. The pipeline also automatically integrated measurements such as streamline density and connectivity from the dMRI data and mapped dMRI data to our template with an isotropic 0.2 mm resolution. It also mapped tracer data to flatmap stacks. Finally, it integrated the 3D image stacks into the Nora-Stackapp and all high-resolution data into the BMCR-Explorer.

## The BMCR-Explorer

The BMCR-Explorer is an online image data viewer that enables visualization of the BMCR data in a high-resolution template space, which is a tremendous advantage for comparative analyses. The viewer shows individual coronal sections of marmoset brain data with an in-plane resolution of 3.0 μm/px. No previous database viewer could show such high-resolution data in a common reference space.

The Explorer includes anterograde tracer image data obtained in 52 marmosets from the Brain/MINDS project [23]. For 19 of these animals, the anterograde tracer data is complemented with retrograde tracer data. All data are accompanied by Nissl and backlit sections.

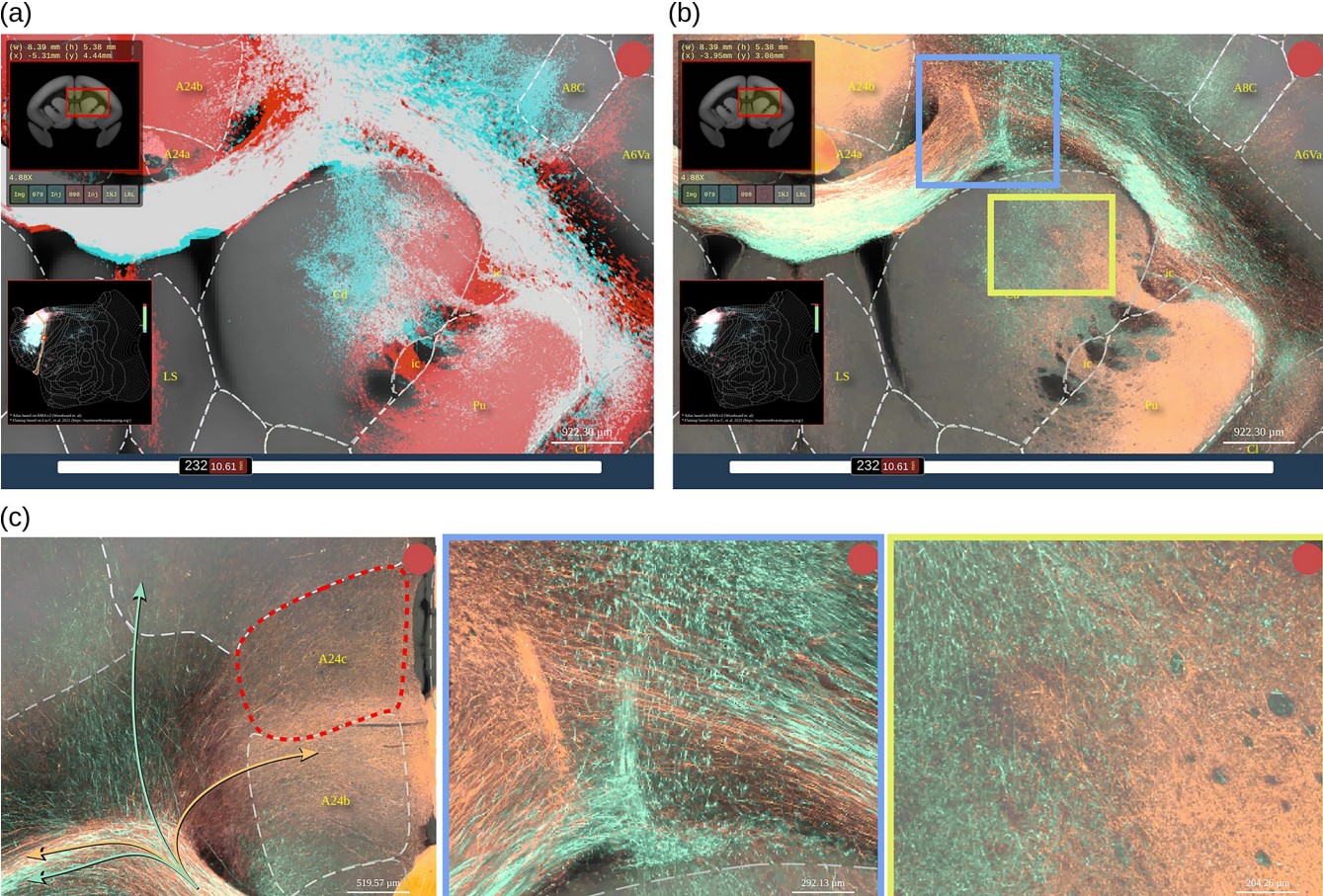

**Fig 3. The BMCR-Explorer.** Example screenshots from the BMCR-Explorer demonstrating a virtual overlay of axon fibers originating from tracer injection into area 24b and area 6M in 2 different marmosets. (a) Tracer segmentation masks for the 2 samples are shown in red and cyan. In this display mode, the overlap between the 2 tracers appears in white. Dashed lines are indicating anatomical annotations from an atlas (in this example, the Brain/MINDS Atlas). The top left and bottom left panels show the position of the ROI in the coronal section and in the flatmap. (b) Same as in panel (a) except that the figure shows the original image data in 2 pseudo colors instead of the segmentation mask. (c) High-resolution views of panel (b) showing fine details of axonal trajectories. Data availability: The images shown in the figure are from 2 marmoset brains with the IDs R01_0098 and R04_0079. The data is publicly accessible from the BMCR-Explorer (http://bmca.riken.jp/). BMCR, Brain/MINDS Marmoset Connectivity Resource.

For each of the 52 injections, a dMRI tracer density image with directional color encoding is included for qualitative comparison with the neural tracer data. The Explorer also incorporates the data from all 145 retrograde tracer injections from the Marmoset Brain Connectivity Atlas [21]. Further, the BMCR-Explorer provides brain annotations for the Brain/MINDS Atlas [20], the Marmoset Brain Connectivity Atlas [21], and the gray and white matter atlases of the Marmoset Brain Mapping project [18,19]. It also includes annotations of major cortical and subcortical regions for the current STPT template [23].

Fig 3 shows examples of anterograde tracer data from 2 different injections. Although the original data are from 2 different marmosets, we can compare them directly in the same high-resolution space for a detailed visualization. In this example, we can see that these axon fibers can intermingle extensively in the white matter (white signal), while being well separated in the striatum and cortex (Fig 3A). Interestingly, axon fibers that seem to be completely mixed in the corpus callosum target different cortical regions when entering the cortex (Fig 3C, left). A major advantage of the BMCR-Explorer is the axonal-level resolution in the coronal plane.

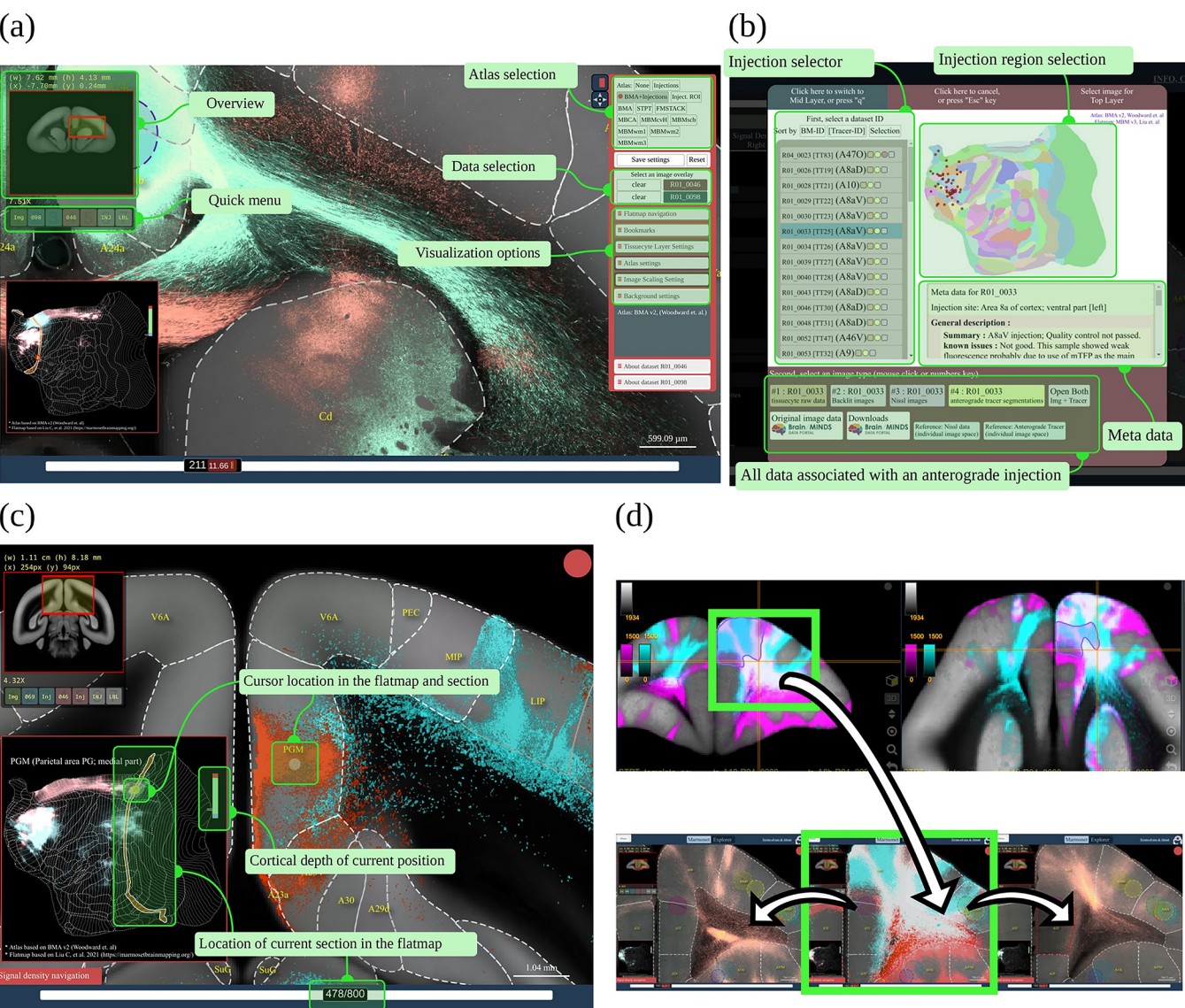

**Fig 4. Interaction between BMCR-Explorer and Nora-StackApp.** Interface of the BMCR-Explorer and the Nora-StackApp. Each example shows 2 anterograde tracers in the BMCR reference space. (a) The BMCR-Explorer shows high-resolution microscopy images of neural tracers from different individuals in a common image space. Panel (b) shows the interface for data selection. (c) The cursor position is shown simultaneously in a cortical flatmap and the current coronal section. (d) The Nora-StackApp viewer can show a number of tracer images simultaneously in 3D that facilitates comparative studies. The viewer supports arbitrary virtual sectioning including sagittal, coronal, or transversal sections and can interact with the BMCR-Explorer. The same location can be opened in high resolution in the BMCR-Explorer. Data availability: Panel (a) shows data from 2 marmoset brains with the IDs R04_0079 and R01_0098, panel (c) shows data from R01_0046 and R01_0098, and panel (d) shows data from R04_0080 and R04_0095. The data is publicly accessible from the BMCR-Explorer (http://bmca.riken.jp/), and Nifti stacks can be downloaded from our repository (https://doi.org/10.60178/cbs.20230630-001). BMCR, Brain/MINDS Marmoset Connectivity Resource.

At high resolution, the different trajectories of axon fibers from 2 samples in the white matter (Fig 3C, middle) or in the striatum (Fig 3C, right) can be discerned, which would only be recognized as mixed at low resolution.

The BMCR-Explorer is equipped with various tools that facilitate the analysis of the anterograde tracer data. Fig 4A shows 2 panels on the top left and bottom left for navigation. They show the position of the ROI within the current brain section and its position in the flatmap, respectively. The panel on the right provides access to various atlas annotations, other datasets

for comparison, or options to adjust visualization parameters such as contrast and opacity. Available data can be listed and selected by choosing an injection site location from a cortical flatmap or by selecting a Brain/MINDS marmoset ID, see Fig 4B. In Fig 4C, the synchronization of flatmap overview and the cross-sectional viewer is shown in more detail. Cortical flatmaps are frequently used for visualizing cortical parcellations and connections. However, due to the nonlinear deformation and flattening, it is difficult to find corresponding locations in the flatmap and in sections of microscopic image data. The BMCR-Explorer has a flatmap viewer that allows the mapping of flatmap locations to high-resolution microscopy images in real time, which makes navigating through a flatmap intuitive and fast.

**The Nora-StackApp.** Although the original datasets consist of high-resolution images, such images are less suitable for offline use and virtual sectioning. The BMCR provides downscaled isometric volume data for offline usage. To support offline exploration, we developed the Nora-StackApp, an image viewer that supplements the BMCR-Explorer with features like virtual sectioning of entire 3D image stacks in a resolution of 100 μm/*vox*. The Nora-StackApp is written in JavaScript and is based on the Nora imaging platform (https://www.nora-imaging.com/). The Nora-StackApp facilitates comparative analysis of marmoset brain image data in 3D. For example, once new image data has been warped to the STPT image space, the Nora StackApp can be used to compare the data to all other data in the BMCR. Also, data aligned to any of the 3 major brain atlases for marmosets can be mapped to the STPT using precomputed warping fields that are part of the BMCR resources. The viewer provides workspaces based on the BMCR annotations, the Brain/MINDS Atlas, and the Marmoset Brain Mapping atlases and can overlay numbers of tracer images and streamline density maps simultaneously. Fig 4D shows a screenshot. The 3D image stacks provide a global picture of the neural architecture in low resolution. At any time, details can be inspected by opening a coronal section in high resolution at the exact same position in the BMCR-Explorer.

## Comparing anterograde neural tracer with dMRI tractography using BMCR data

Diffusion MRI is widely used for studying primate brain connectivity in vivo. It is thought to reflect the anisotropy of axonal fiber structures. However, the estimates are imperfect [30,31]. The limitations of dMRI tractography, such as false positives, false negatives, and the inability to determine the directionality of connections, have been well documented in the literature [32,33]. Moreover, a study combining anterograde tracer experiments with dMRI tractography in macaques revealed that a challenge for dMRI tractography is to penetrate superficial white matter systems to reach deep white matter, which may affect the accurate estimation of intercortical connections [34].

The BMCR provides anterograde tracer data showing axonal projections from the injection site. When combined with dMRI, it can be used as a "ground truth" for comparison with dMRI-based tractography, as has been previously proposed in studies involving the macaque brain using retrograde or anterograde tracer data [34,35].

We generated a population dMRI dataset based on 23 individual scans. For the comparison, we used streamlines generated from the population average image. We compared streamline and tracer density maps from areas A32 and A8aV, 2 distinct regions in the PFC where our tracer signals show non-overlapping pathways. Interestingly, a comparison of the 2 pathways showed that they are also separated in the corpus callosum and the internal capsule. The spatial gap between the 2 injection sites, and the non-overlapping pathways in close proximity, make these 2 regions a perfect example for comparison with dMRI fiber tractography.

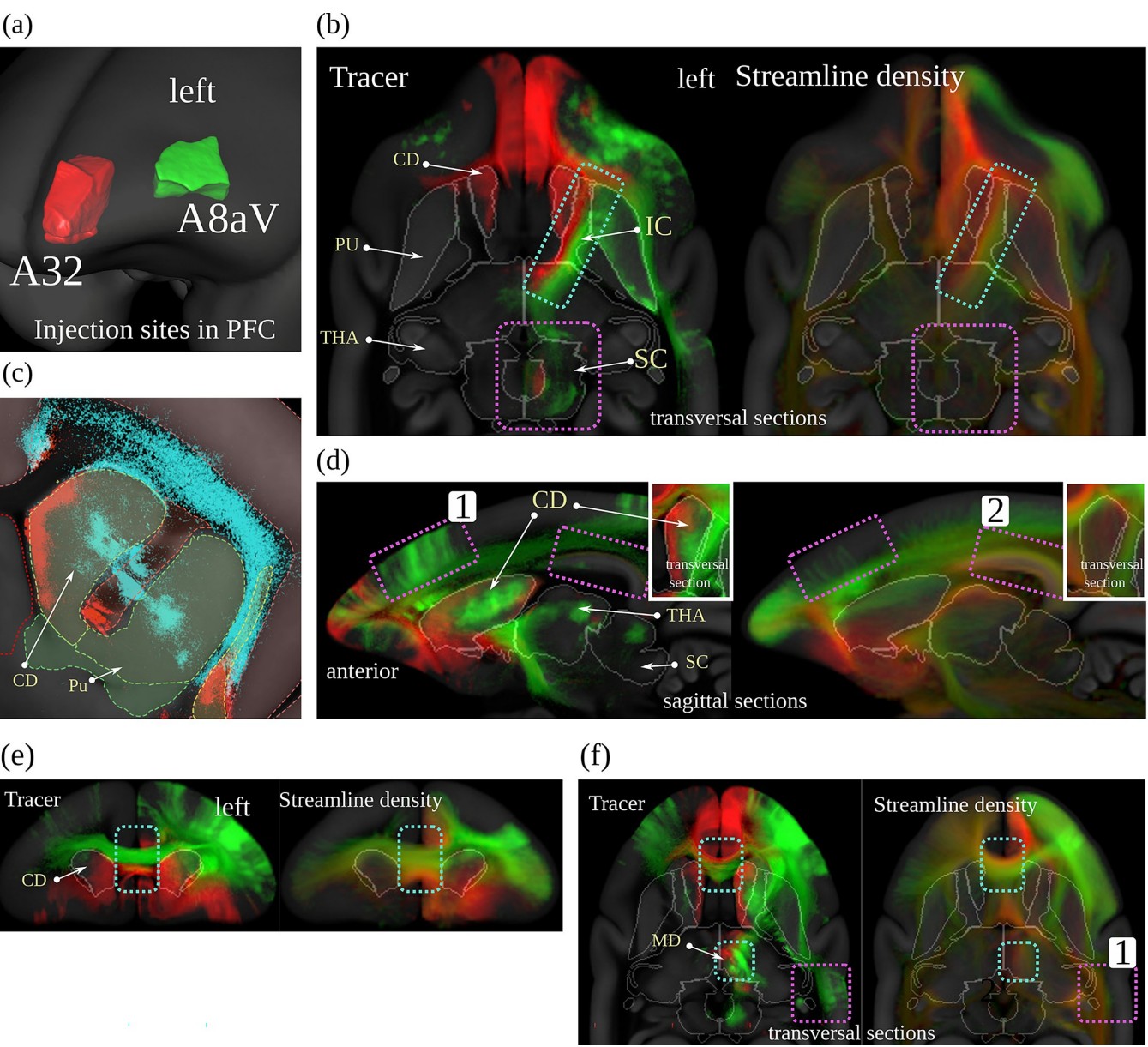

**Fig 5. dMRI tractography and neural tracer.** Visual comparison between dMRI-based streamline density and anterograde neural tracers originating in 2 distinct regions in the marmoset PFC. Each of the 2 colors represents the maximum intensity over all anterograde tracer images that have been injected into one of the 2 regions in the marmoset PFC (A32: red, A8aV: green (cyan in (c))). Panel (a) shows the injection regions, (b) shows separate streams in the internal capsule. Panel (c) shows details in the caudate nucleus. In panel (d), cortical projections differ in tracer and dMRI (purple boxes). Panel (e) shows separate streams in the corpus callosum, and in (f) we see wrong streamlines (red) in dMRI (purple boxes). The similarity between the images suggests that dMRI reflects real brain connectivity (cyan boxes with dashed borders in (b), (d), (e), and (f)) but also shows evidence of the relative imprecision of dMRI data in terms of specificity and sensitivity (violet boxes with dashed border). Data availability: The source code that generated the image is publicly available (https://doi.org/10.5281/zenodo.7906530, filename: BMCR_Fig 05.m), and the corresponding data are publicly available on the CBS Data Sharing Platform (https://doi.org/10.60178/cbs.20230630-001). dMRI, diffusion-weighted MRI; PFC, prefrontal cortex.

Figs 5 and 6 compare the tracer signal to the streamline density maps generated from dMRI fiber tractography. As we have multiple injections into both regions, we combined the tracer signal of all injections at each site by first normalizing the tracer signal intensity for each image and then taking the maximum value across all samples. Further details regarding the tracing and the fiber tractography method are explained in the Methods section.

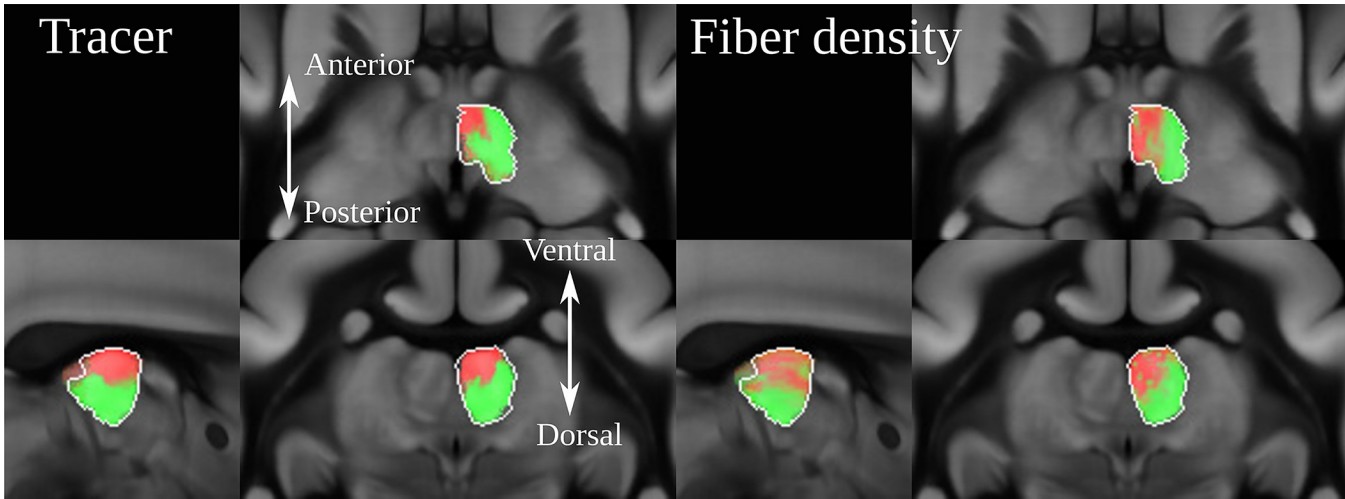

**Fig 6. The image shows the strong projections derived from anterograde tracing and dMRI in the mediodorsal thalamic nucleus.** Both dMRI and anterograde tracer suggest that A32 projects anteromedially while A8aV projects posterolaterally. dMRI, diffusion-weighted MRI.

We found remarkable similarities between the dMRI and tracer data. Fig 5B shows that both tracer and streamlines pass the internal capsule in segregate streams. The tracts project strongly into the mediodorsal thalamic nucleus (MD). There has been strong evidence from dMRI tractography that A32 projects anteromedially while A8aV projects posterolaterally [36], which could be confirmed in our comparison of tracer and dMRI tractography (Fig 5F and Fig 6). Tracts from both regions also pass the corpus callosum in separate streams (Fig 5E and 5F). The tract from A8aV runs ventroposteriorly in the corpus callosum, while the tract originating in A32 runs ventralposteriorly.

However, there are also clear discrepancies between the 2 sets of results. Fig 5C shows diffuse tracer projections into the caudate nucleus (CD). The image shows a close-up view of the segmented tracer of 2 examples, one for each injection site. A32 projects medially and A8aV laterally. These projections are not present in the dMRI tractogram. Similarly, we can observe thin, diffused connections from the thalamus projecting strongly into the superior colliculus (Fig 5B), which are also absent in the dMRI data. It is also occasionally difficult to reproduce cortical projections correctly. For example, the region highlighted by box 1 in Fig 5D shows strong cortical projections in the tracer signal, which are underrepresented in the data derived from dMRI. Conversely, box 2 shows strong cortical projections in dMRI that are not present in the tracer data. A similar observation can be made in box 1 in Fig 5F, where dMRI tractography suggests connections which are not supported by the tracer data.

These direct comparisons add to the evidence of the relative imprecision of dMRI data in terms of specificity and sensitivity, as previously proposed based on tracer and tractography data comparisons in the macaque brain [30]. This highlights the need for ground truth provided by cellular-resolution tracers. However, available anterograde tracer data for nonhuman primates, as used in [34], is sparse. Our study marks the first time a large dataset of tracer data from nonhuman primate brains has been made publicly available alongside high-quality dMRI measurements in the same image space. Such data are important to further study how fiber tracking techniques and their parameters affect the comparison with tracer data as well as their influence on the resulting connectomes [33,37]. Moreover, these data can be used to improve tractography accuracy when incorporated as anatomical priors [38].

### Integrating retrograde tracer data of the Marmoset Brain Connectivity Atlas

The Marmoset Brain Connectivity Atlas (https://www.marmosetbrain.org/) provides post-processed image data for 145 retrograde tracer injections. Retrograde tracers can reveal back-projection, making them a valuable counterpart to our BMCR anterograde tracer data, given the fact that most corticocortical connections are reciproical [39]. The data includes the locations of cell bodies in the Paxinos stereotaxic reference space [40]. We mapped all 145 datasets to our BMCR template image space. The matrix in Fig 7B shows the normalized cross-correlation of the anterograde tracer signal and density of retrogradely labelled cells in the cortex between pairs of flatmap stack data. Data were paired with respect to the closest injection site distance with respect to the STPT template space. Fig 7A shows the similarity of the antero-grade tracer data as a reference. Fig 8 shows examples of the images as maximum intensity projections.

Fig 7C shows similarity matrices for the anterograde and retrograde tracers, respectively. Both matrices positively correlate (Pearson correlation coefficient of 0.61, random permutation test with $n$ = 1 M repetitions gave a $P$-value of $<10^{-6}$; we only considered the upper triangle of the symmetric matrices).

The mapped data were integrated into the BMCR-Explorer and the Nora-StackApp. We also mapped cell density onto flatmap stacks. The data suggest that anterograde and retrograde tracers exhibit similar projection patterns, but also reveal important differences in their laminar patterns, which are essential for defining feedforward and feedback connection patterns [41,42]. The similarity between the tracers suggests that the retrograde data from the Marmoset Brain Connectivity Atlas are well aligned with our STPT template image space. Fig 7D shows an example in the BMCR-Explorer for 2 nearby injections in the PFC. It illustrates an example of remarkable spatial correspondence between the 2 tracer patterns.

## Discussion

The work in this paper is part of Japan's Brain/MINDS project [43–45]. The project is working on the construction of an integrated, multiscale structural map of the marmoset brain from data acquired using several imaging modalities such as two-photon imaging, in situ hybridization [46], and dMRI. The BMCR tools described here, allow exploration of the first publicly available multimodal dataset of anterograde tracer injections in a primate brain. The integration of neuroanatomical tracers with structural MRI allows the user to navigate bidirectionally between macroscopic anatomical information obtained by MRI and cellular-level neuroanatomical information obtained by tracers and histological techniques. In addition, the BMCR allows direct comparisons between anterograde and retrograde tracer injection data, due to the integration of datasets from the Marmoset Brain Connectivity Atlas [47]. While our focus here is on the connectivity of the marmoset PFC, we are currently working on expanding the data with anterograde tracer injections into other regions of the cortex. Other planned features include data from disease model marmosets.

### Relation to previous work

The development of the BMCR is part of the international trend towards open-access resources for the exploration of brain connectivity. Connectivity datasets into multimodal platforms has recently been identified as a priority area for the advancement of translational neuroscience [48], and the present resource addresses this need. In this regard, the BMCR extends and complements capabilities offered by other online resources. For example, the Human

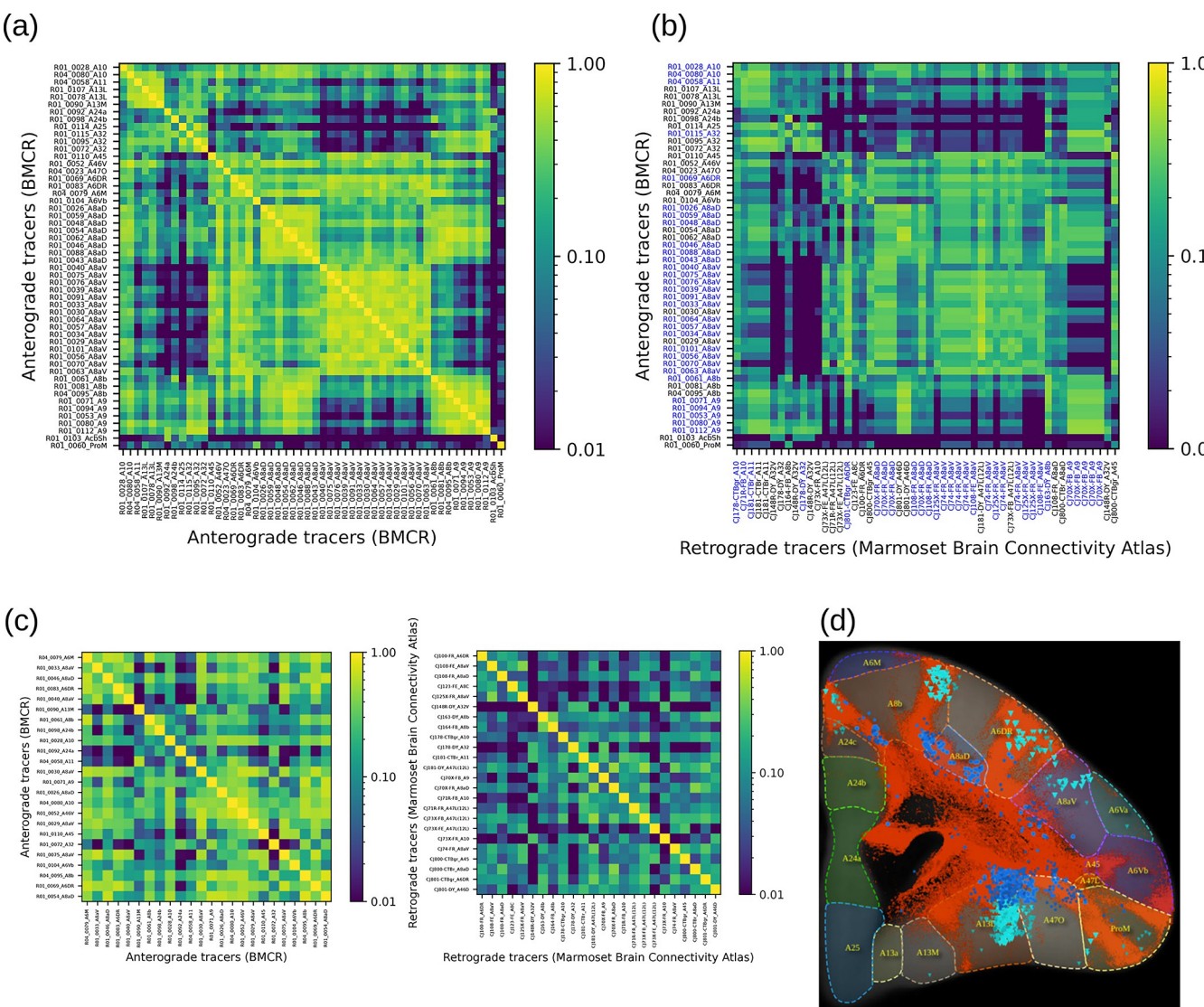

**Fig 7. Integration of the Marmoset Brain Connectivity Atlas data.** We integrated retrograde tracer data from the Marmoset Brain Connectivity Atlas into the BMCR. The top matrices show the visual similarity of tracer data in the cortex of marmosets (based on normalized cross-correlation). (a) Similarity between all pairs of anterograde tracers as a reference, and for (b), pairs were formed between BMCR anterograde tracers and retrograde tracers from the Marmoset Brain Connectivity Atlas based on the distance of the nearest injection site. Panel (c) shows the similarity among tracer signals injected into different locations in the cortex for the 2 tracer types. Both matrices suggest similar connectivity patterns (Person correlation with a *P*-value of $<10^{-6}$). Panel (d) shows an example of notable overlap between a pair of datasets from both projects with similar injection sites. The retrograde signal is shown over the segmentation of the anterograde tracer (red). Data availability: The source code that generated the plots is publicly available (https://doi.org/10.5281/zenodo.7906530, filename: BMCR_Fig 07_Fig 08.ipynb), and the corresponding data are publicly available on the CBS Data Sharing Platform (https://doi.org/10.60178/cbs.20230630-001). BMCR, Brain/MINDS Marmoset Connectivity Resource.

Connectome Project compiles an extensive amount of such structural and functional neural data of the human brain [49]. However, the acquisition of large-scale structural connectivity data is limited to dMRI imaging techniques. For animal models, tracer techniques are frequently used to map neural connectivity in more detail [50,51], with the Allen Mouse Brain Connectivity Atlas [1,2] and the Marmoset Brain Connectivity Atlas [21,22] offering 2 examples where the results of a large number of tracer injections is made publicly available, and accompanied by an average template brain, brain annotations, and tools for visualization and exploration. The Allen Mouse Brain Connectivity Atlas provides anterograde tracer data in the

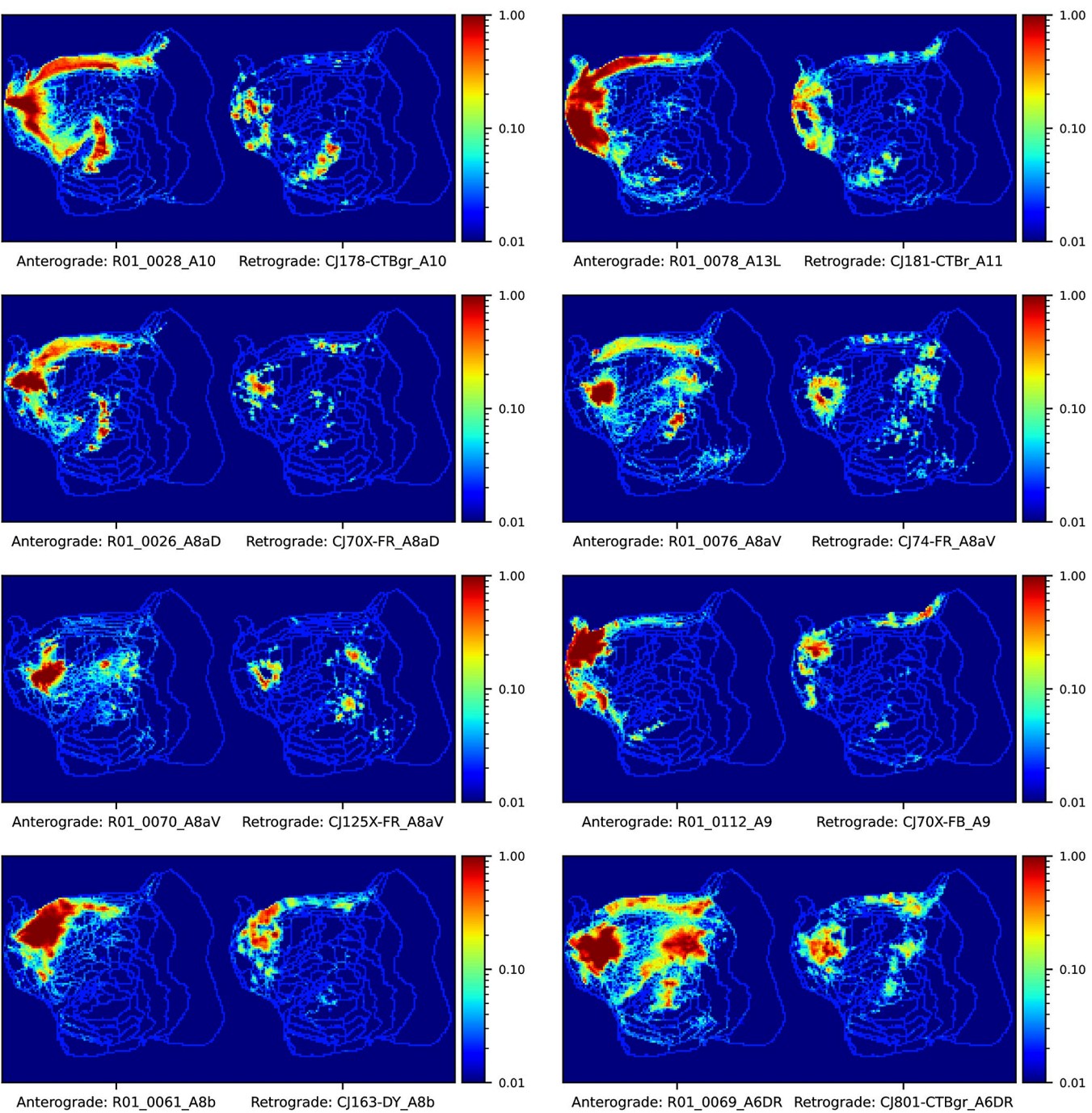

**Fig 8. Visual comparison between anterograde and retrograde tracers.** We measured the similarity between anterograde and retrograde tracers based on the normalized cross correlation of flatmap stack images of tracer patterns in the cortex. The image shows maximum intensity projections from pairs of images of anterograde and retrograde injections from the BMCR and the Marmoset Brain Connectivity Atlas, respectively. The pairs were formed with respect to the closed distances between injection sites. The shown flatmap shapes are consistent with the Marmoset Brain Mapping flatmap. The outline of cortical regions are based on annotations from the Marmoset Brain Connectivity Atlas. Data availability: The source code that generated the figures is publicly available (https://doi.org/10.5281/zenodo.7906530, filename: BMCR_Fig 07_Fig 08.ipynb), and the corresponding data are publicly available on the CBS Data Sharing Platform (https://doi.org/10.60178/cbs.20230630-001). BMCR, Brain/MINDS Marmoset Connectivity Resource.

mouse brain, which have been acquired with an STPT system (similar to the present BMCR). In contrast, the Marmoset Brain Connectivity Atlas reconstructs data from cortical retrograde tracer injections from histological sections of the marmoset brain, followed by 3D reconstruction and registration to stereotaxic reference space.

Overall, there are still only a few integrative tracer databases, to some extent due to the fact that the systematic mapping, processing, and visualization of imaging data are labor-intensive and costly. Alternative approaches, such as the CoCoMac project [52,53], aim at accumulating and integrating the output of various research studies to better understand global brain connectivity. However, this relies on heterogeneous data sources and lacks access to ready-to-use image data.

## Integration of different modalities of structural connectivity

Diffusion MRI is currently the most widely used technique for studying brain connectivity. dMRI provides a key link to neuropsychological and neurosurgical practice, in particular, due to its in vivo applicability. The main image features of dMRI are based on relatively large and oriented axonal fiber bundles, which create an anisotropy of the dMRI signal. However, estimates of true structural connectivity based on this technique are imperfect [30,31], depend to a large extent on post-processing steps [54], and do not allow estimation of the direction of information flow. Anterograde injection studies are important, because they can be used to validate connectivity measures based on diffusion MRI (HARDI) [55,56]. The BMCR integrates a HARDI population image to foster the comparison between tracer and structural dMRI [31,37,57]. In this context, the BMCR provides a rich platform to enable future studies aimed at refining and validating dMRI, by providing simultaneous ground truth datasets, and access to histological information. Fig 5 shows that the similarities in topology between tracer injection patterns and dMRI streamline densities can be remarkable. Although the bundles of different injections run fairly close to each other, the dMRI streamline densities stay consistently apart, which suggests that assumptions about topological preservation, which most tractographic approaches rely on, are generally valid. However, there are important differences and the cellular tracer data allow estimates of the directionality of the connection, which is not recoverable from dMRI data.

Further, the BMCR integrates anterograde tracer data with retrograde tracer data. Our recent study has shown that signals from combined anterograde and retrograde tracer injections correlate well in the PFC, suggesting a strong correlation between projection patterns and back projection patterns [23]. However, there are consistent reports of non-reciprocal pathways in both the macaque and marmoset brain [41], and the combination of both tracer modalities is the best approach to further investigate this issue. In addition, the laminar patterns of both cell bodies and terminals are critical for establishing patterns of hierarchical flow of information in multi-areal pathways [58]. To encourage further investigation of such relationships, we supplemented our data with the large set of retrograde data from the Marmoset Brain Connectivity Atlas. Our preliminary comparison with the Marmoset Brain Connectivity Atlas data supports our observation of spatially correlated tracer patterns and reveals different laminar patterns for cell bodies and terminals within cortical columns.

## Materials and methods

### The BMCR image dataset

This section briefly describes image acquisition. Details regarding the neural tracer and the acquisition can be found in the methods of the previous report [23]. Tables 2 and 3 list

**Table 2. Marmoset details.**

|  | ID | Sex | Age | Injection | A/N/B | dMRI | R | V |
|---|---|---|---|---|---|---|---|---|
| 1 | R01_0026 | F | 5.0Y | A8aD | x |  |  | x |
| 2 | R01_0028 | M | 5.1Y | A10 | x | x |  |  |
| 3 | R01_0029 | F | 5.0Y | A8aV | x | x |  | x |
| 4 | R01_0030 | F | 5.3Y | A8aV | x | x |  |  |
| 5 | R01_0033 | F | 5.3Y | A8aV | x | x |  | x |
| 6 | R01_0034 | F | 2.8Y | A8aV | x | x |  |  |
| 7 | R01_0039 | F | 5.2Y | A8aV | x | x |  | x |
| 8 | R01_0040 | M | 5.6Y | A8aV | x | x |  |  |
| 9 | R01_0043 | F | 3.3Y | A8aD | x | x |  | x |
| 10 | R01_0046 | M | 3.4Y | A8aD | x | x |  |  |
| 11 | R01_0048 | F | 2.4Y | A8aD | x |  |  | x |
| 12 | R01_0052 | F | 6.2Y | A46V | x | x |  |  |
| 13 | R01_0053 | F | 7.0Y | A9 | x | x |  | x |
| 14 | R01_0054 | F | 8.1Y | A8aD | x | x |  |  |
| 15 | R01_0056 | F | 4.5Y | A8aV | x |  |  | x |
| 16 | R01_0057 | F | 4.1Y | A8aV | x | x |  |  |
| 17 | R01_0059 | M | 8.1Y | A8aD | x |  |  | x |
| 18 | R01_0060 | F | 7.3Y | ProM | x |  |  |  |
| 19 | R01_0061 | M | 8.5Y | A8b | x |  |  | x |
| 20 | R01_0062 | M | 8.5Y | A8aD | x | x |  |  |
| 21 | R01_0063 | M | 8.3Y | A8aV | x | x |  | x |
| 22 | R01_0064 | M | 8.3Y | A8aV | x | x |  |  |
| 23 | R01_0069 | F | 6.6Y | A6DR | x |  |  | x |
| 24 | R01_0070 | F | 6.6Y | A8aV | x |  |  |  |
| 25 | R01_0071 | M | 7.8Y | A9 | x |  |  | x |
| 26 | R01_0072 | F | 10.4Y | A32 | x |  |  |  |

"A/N/B" means Anterograde tracer, Nissl and backlit images, "R" means retrograde tracer data and "V" means manual landmark and injection site annotations were generated and were used for validation.

dMRI, diffusion-weighted MRI.

individual marmoset data, including injection site locations. Fig 9 shows the injection site locations in the STPT image template. Auxiliary resources are listed in Table 4.

The dataset comprises the multimodal image data of 52 individuals with 52 anterograde tracer injections. The injections were placed into 21 disjunctive brain regions in the left hemisphere of the marmoset PFC. Fig 9 shows the location of the 52 injections. The acquisition took place in 5 steps. The anterograde TET-amplified AAV fluorescent neural tracer contained a mixture of clover and 1/4 amount of presynapse targeting mTFP1. For 19 brains, the tracer was injected in a mixture with AAV2retro-EF1-Cre, a retrograde tracer.

First, postmortem, an ex vivo full-brain dMRI was imaged with a 9.4 Tesla MRI animal scanner (Bruker Optik GmbH, Germany). Animals were perfusion-fixed using 4% paraformaldehyde (PFA), and their brains were extracted for ex vivo imaging. During this process, brains were encased in a sponge and submerged in a fluorine solution within a plastic container to prevent MRI interference. Vacuum degassing minimized artifacts. T2-weighted (T2W) images were taken with a spatial resolution of 100 μm × 100 μm × 200 μm (scan time 3 h, 20 min). Afterwards, dMRI images based on the HARDI protocol were acquired, with b-values of 1,000, 3,000, and 5,000 $s$/mm$^2$ with an isotropic image resolution of 0.2 mm and 128

**Table 3. Marmoset details (continued).**

| | ID | Sex | Age | Injection | A/N/B | dMRI | R | V |
|---|---|---|---|---|---|---|---|---|
| 27 | R01_0075 | F | 5.6Y | A8aV | x | | | x |
| 28 | R01_0076 | M | 8.9Y | A8aV | x | x | | |
| 29 | R01_0078 | F | 8.9Y | A13L | x | | | x |
| 30 | R01_0080 | F | 7.4Y | A9 | x | | | |
| 31 | R01_0081 | F | 7.6Y | A8b | x | | | x |
| 32 | R01_0083 | F | 6.0Y | A6DR | x | x | | |
| 33 | R01_0088 | F | 3.7Y | A8aD | x | x | x | x |
| 34 | R01_0090 | F | 2.9Y | A13M | x | x | x | |
| 35 | R01_0091 | M | 2.4Y | A8aV | x | x | x | x |
| 36 | R01_0092 | M | 2.4Y | A24a | x | | x | |
| 37 | R01_0094 | M | 10.8Y | A9 | x | x | x | x |
| 38 | R01_0095 | M | 2.3Y | A32 | x | | x | |
| 39 | R01_0098 | M | 3.3Y | A24b | x | | x | x |
| 40 | R01_0101 | F | 9.2Y | A8aV | x | | | |
| 41 | R01_0103 | M | 8.2Y | AcbSh | x | | x | x |
| 42 | R01_0104 | M | 5.3Y | A6Vb | x | | x | |
| 43 | R01_0107 | M | 2.5Y | A13L | x | | x | x |
| 44 | R01_0110 | F | 6.7Y | A45 | x | | x | |
| 45 | R01_0112 | F | 6.0Y | A9 | x | | x | x |
| 46 | R01_0114 | F | 9.8Y | A25 | x | | x | |
| 47 | R01_0115 | F | 8.4Y | A32 | x | | x | x |
| 48 | R04_0023 | M | 2.6Y | A47O | x | | x | |
| 49 | R04_0058 | F | 5.0Y | A11 | x | | x | x |
| 50 | R04_0079 | F | 4.3Y | A6M | x | | x | |
| 51 | R04_0080 | F | 3.2Y | A10 | x | x | x | x |
| 52 | R04_0095 | F | 2.6Y | A8b | x | | x | |

"A/N/B" means Anterograde tracer, Nissl and backlit images, "R" means retrograde tracer data and "V" means manual landmark and injection site annotations were generated and were used for validation.
dMRI, diffusion-weighted MRI.

independent diffusion directions (scan time 6 h, 39 min). We followed Bruker's standard recommended settings for the diffusion directions during imaging [59]. Further details regarding the dMRI acquisition protocol can be found in [60].

Next, the entire brain with the fluorescent anterograde tracer signal was imaged by fully automated Tissuecyte 1000 and Tissuecyte 1100 STPT (TissueVision, Cambridge, Massachusetts, USA). For STPT, the entire brain was embedded into agarose and mounted under the microscope. After imaging the block face, automated vibratome sectioning was performed. These steps were repeated automatically until the entire brain was imaged. The coronal in-plane image resolution of the raw data was 1.385 μm × 1.339 μm$^2$ with in total of about 19,000 × 16,000 pixels. The image of each section contained 3 channels: the first channel primarily represents the auto-fluorescent background (i.e., the entire brain structure), while the second channel captures the tracer signal. There was a large difference in tracer intensity within and outside the injection site. To capture weak tracer signals, the dynamic range was sacrificed in the injection site such that the signal in the injection site was saturated in the first 2 channels. For compensation, the third channel was used to represent the details (the infected cell bodies) in the injection site. Fig 10A shows an example.

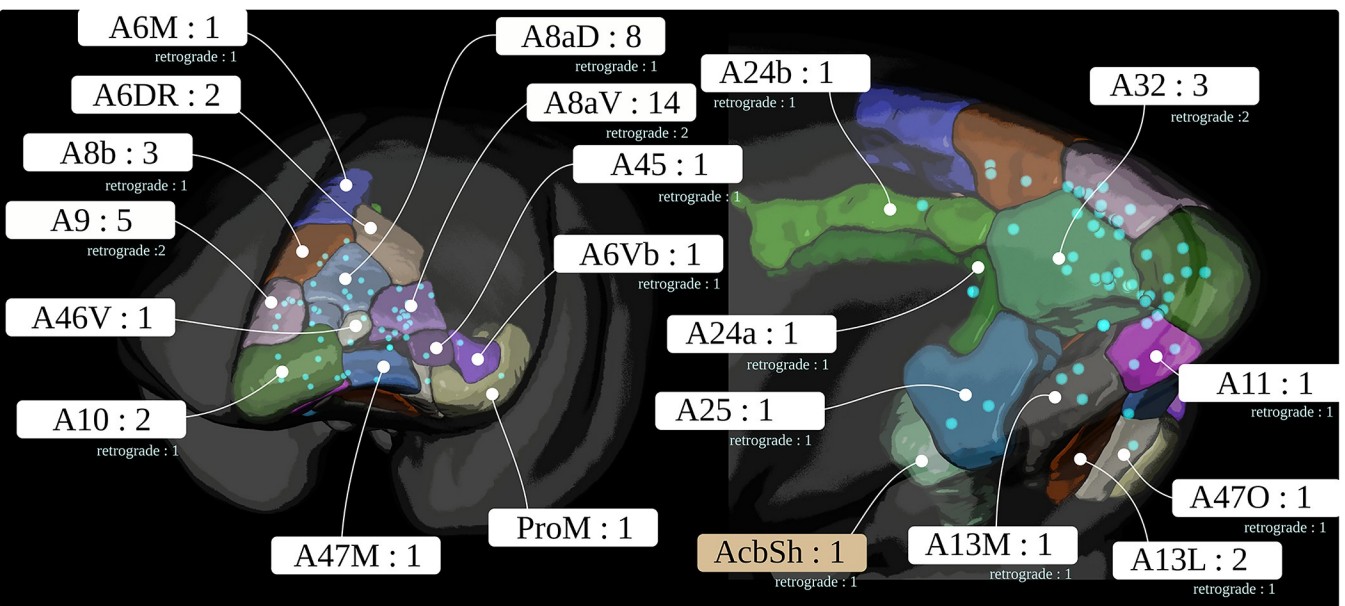

**Fig 9. Injection sites.** The figure shows the locations of all 52 anterograde and corresponding 19 retrograde tracer injections in the marmoset PFC. PFC, prefrontal cortex.

Every 10th section, which corresponds to a 500 μm offset, was recovered and fluorescently immunostained for Cre. Fluorescent images were captured with an all-in-one microscope (Keyence BZ-X710). The in-plane resolution was 3.774 μm/px. The images contain 2 color channels. The first channel contains the signal of the retrograde tracer signal (cell bodies). In the second channel, the anterograde tracer signal was also captured by the Tissuecyte microscope. In addition, the second set of slices, also in a 10-section interval, was collected. The sections were imaged twice before (backlit) and after Nissl staining with the same microscope (Keyence BZ-X710) with a pixel resolution of 3.774 μm/px.

**Table 4. Auxiliary resources.**

| Resource | Description |
|---|---|
| Transformation between STPT and the Brain/MINDS Atlas templates | ANTs registration warp-field |
| Transformation between STPT and Marmoset Brain Mapping atlases v2 and v3 templates | ANTs registration warp-field |
| Transformation between STPT and BMCR templates | ANTs registration warp-field |
| Transformation between STPT and Flatmap stack (Brain/MINDS Atlas) | ANTs registration warp-field |
| Transformation between STPT and Flatmap stack (Marmoset Brain Mapping atlas) | ANTs registration warp-field |
| 26×3 manual injection site masks | Created by three experts, used for data validation |
| 20 landmarks in the STPT image space | Used for data validation |
| 20×3 landmarks for 26 datasets | Created by 3 experts, used for data validation |
| Transformation between marmoset and human MNI image space | ANTs registration warp-field |

BMCR, Brain/MINDS Marmoset Connectivity Resource; STPT, serial two-photon tomography.

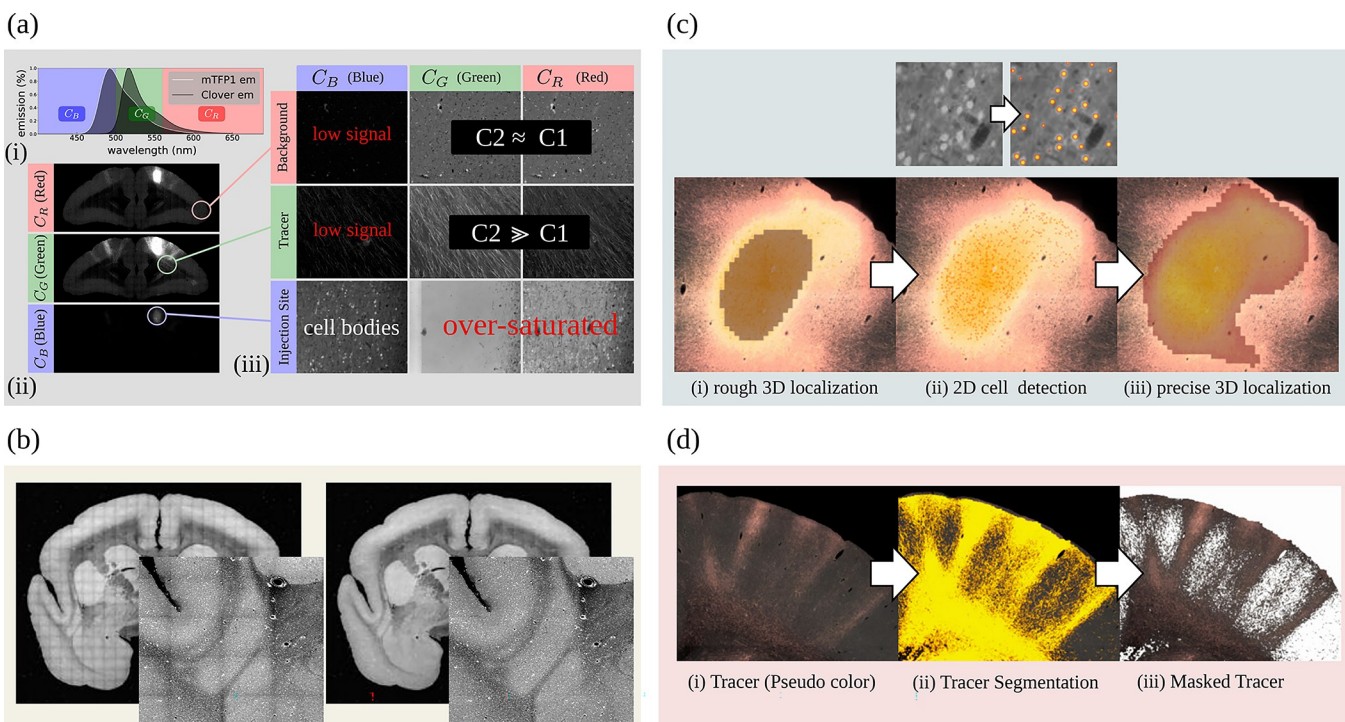

**Fig 10. Anterograde tracer data from STPT.** (a) The fluorescent emission profile of the anterograde tracer. (b) A coronal section of the STPT microscope (first channel) before and after intensity correction. (c) Illustration of the injection sites localization and (d) the anterograde tracer segmentation. STPT, serial two-photon tomography.

## Marmoset experiments

The animals were individually housed in stainless steel cages,which were cleaned with water and dried every weekday morning. They had ad libitum access to tap water and 40 g/individual/day food pellets (CMS-1M; CLEA Japan, Tokyo, Japan), supplied each day before noon. For animal enrichment, a piece of castella cake was given once a day as a snack at 15:00 h. The animal rooms were controlled at 28 ± 1˚C and 50 ± 20% humidity under a 12 h lighting schedule (light from 8:00 to 20:00) and were regularly tested for *Salmonella* spp. and *Shigella* spp. using the culture method. The animal care staff conducted daily health observations, which include monitoring the fecal consistency, presence or absence of vomiting, urinary characteristics, food intake, posture, activity level, facial expression, presence or absence of injuries or bleeding, and the quality of the animal's fur coat. Surgery for tracer injections was performed as previously described in [61] under deep anesthesia induced by isoflurane (2% to 3%) inhalation. Meloxicam (0.2 mg/kg) was used as an analgesic after the surgery. The animals were killed 4 weeks after tracer injection by transcardial perfusion with 4% PFA in 0.1 M phosphate buffer (pH 7.4), and the brain was retrieved for ex vivo MRI and STPT imaging. We euthanized the animals using an overdose of Pentobarbital or Sodium thiopental when it was necessary to kill them for perfusion or due to illness.

## The processing pipeline

This section describes details regarding all post-processing methods. Fig 2 outlines the processing pipeline. The image post-processing part of the pipeline works in a fully automated manner and does not require manual interaction. The pipeline was written in a mixture of python

3 (www.python.org) and Matlab (MathWorks) code. Each pipeline step was executed by a dedicated script running on an Ubuntu Linux cluster system. The entire pipeline was orchestrated by a Python-based pipeline database system that kept track of data dependencies and facilitated the parallel launching of scripts using the SLURM workload manager (www.schedmd. com).

All image data has been aligned to our BMCR template space, a left-right symmetric population average template of a marmoset brain with an isotropic resolution of 50 μm. We chose a symmetric template because all injections were placed in the left hemisphere. Averaging the left and right hemispheres allowed us to double the number of samples and minimize potential biases. Additionally, the symmetric template streamlined the annotation of brain structures and facilitated the integration with the Marmoset Brain Connectivity Atlas, which provides data for only 1 hemisphere.

For image alignment, we used the ANTs image registration. If not stated otherwise, we used a multiscale affine image registration followed by a multiscale deformable SyN registration [29] with normalized mutual information as a metric.

**STPT image stitching.** The image stitching was done in Matlab based on in-house code. The TissueCyte microscope generates a large number of image tiles. The size of each image tile has been set to 720×720 pixels with a spatial in-plane resolution of 1.385 μm×1.339 μm$^2$. The tiles were provided as 16 bit tiff files. The microscope outputs the offset for each image tile in plane-text as 3D world coordinates with micrometer resolution. The coordinates are sufficiently precise to allow the reconstruction of an entire image section by aligning and fusing all image tiles according to their world coordinates. We set a small overlap between adjacent image tiles (about 80 pixels) and cropped 50 pixels from the image tile boundaries before fusing them using a linear blending function. The size of an entire brain section was approx. $19,000 \times 16,000$ pixels.

Distortions in the microscope's optical path created an inhomogeneous vignetting effect in the tile images. Hence, before stitching, we applied an intensity correction. We estimated the shading field by averaging over a large set of image tile samples and divided each tile by the result. Intensity correction for tile images was only applied to the first 2 channels (background and tracer). The third channel, which has a clear signal around the injection site but a low contrast and a bad signal-to-noise ratio elsewhere was excluded from the correction due to the small number of tile samples with meaningful content. Further details regarding the correction algorithm can be found in our technical report [62]. An example before and after stitching is shown in Fig 10B.

For all 3 image channels, we created 3D image stacks with an isotropic image resolution of 50 μm. The image stacks were saved in the NIfTI file format (https://nifti.nimh.nih.gov/). The full resolution image sections were stored as 16 bit lossless PNG images.

**Injection site location.** The pipeline locates the injection site in 2 steps. Fig 10C shows an example. First, we located its rough position as the brightest connected structure in the 3D image stack of the third STPT color channel. In that channel, the cells in the injection site appear bright, while there is almost no signal outside the injection site. We utilized Matlab for localization. We applied Gaussian smoothing, followed by the application of an intensity threshold (half the maximum intensity in the image) and connected component analysis. We determined the volume of the injection site as the largest connected component.

In the second step, an artificial neural network analyzed all full-resolution 2D STPT brain sections to identify infected cell bodies. To speed up the process, the screening only took place for that part of the 2D image sections that intersected with the volume of the injection site, which was determined in the first step. We used a large margin to make sure that all parts of the injection site were included. The network architecture was a 2D U-Net [28], a

convolutional neural network for biomedical image processing. We trained the network on images with 512×512 pixels to map STPT images to probability maps for the locations of the centers of cell bodies. Local maxima in the probability maps that exceeded a probability of 0.5 were considered cell locations.

We created a training dataset with images of 6,068 manually annotated cell bodies that appeared in 44 different 2D images. We selected the images from 10 different marmoset brains. The U-Net was based on the original implementation, with a depth of 4. Most upper layers had 64 features after the first convolution layer. The number doubled after each pooling operation to up to 512 features. We further used drop-out and batch normalization [63,64]. The training procedure included augmentation of the image with deformations, as well as changes in intensity and contrast. Details regarding the architecture and a performance evaluation can be found in our technical report [62].

**Anterograde tracer segmentation.** This step takes the raw STPT image sections as input and segments the anterograde tracer signal from the background. This was done by applying a 2D U-Net to the data. The network takes image patches combining the first 2 STPT image channels as input and returned a tracer signal probability map as output. Both input channels show the auto-fluorescent background, but the neural tracer signal was significantly brighter in the second channel. The difference helped the network to better distinguish the tracer-positive pixels from the background. The pipeline applied the network to all image sections. Fig 10D shows an example.

The pipeline generated 2 kinds of image data from the segmented data. The anterograde tracer density and the normalized anterograde tracer signal intensity. The tracer *density* is a 3D image stack in which voxels represent the amount of tracer positive pixels in a 50×50 μm$^2$ area in a raw 2D image section. The normalized tracer signal *intensity* takes the actual tracer intensity into account. The neural tracer is much brighter in the second channel than in the first channel, while the background appears similarly bright in both channels. We obtained the signal intensity by subtracting the first channel from the second channel. We then normalized the intensity with respect to its strongest signal outside the injection site. The injection site itself was excluded from the calculation because of the saturated signal in the injection site, and thus values within the injection site volume did not represent a meaningful quantity for normalization. Fig 11 shows an example of 3D reconstruction of tracer density. The BMCR provides 2 kinds of tracer segmentations.

The data for training the U-Net were generated in a semi-supervised way. We applied a threshold to the tracer intensity to generate a large set of labeled brain image sections. We

(a) (b) (c)

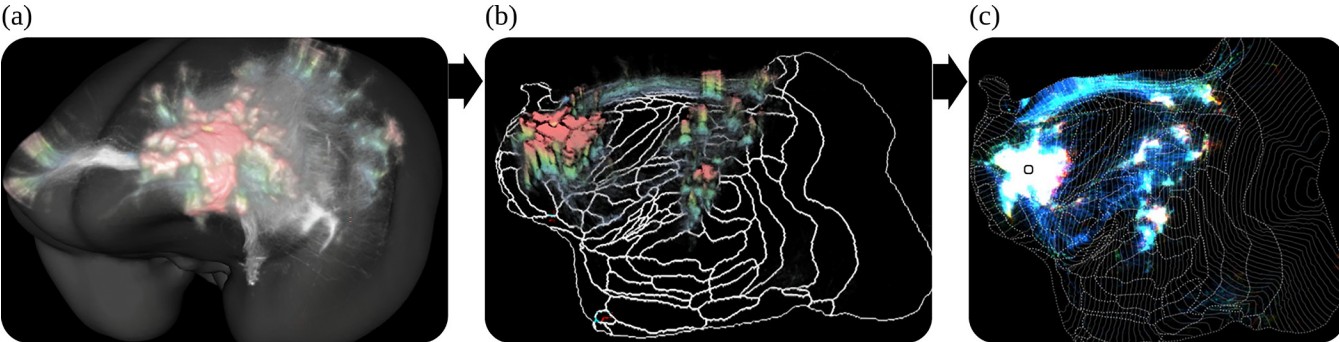

**Fig 11. A 3D reconstruction of an anterograde tracer density.** The intensities in the cortex have been colored according to cortical depth. From left to right: (a) The signal in the STPT template image space, (b) the signal in the left hemisphere mapped to a 3D flatmap stack, and (c) the 2D projection of the flatmap stack. STPT, serial two-photon tomography.

manually screened and selected about 600 image sections from 20 different marmoset brain image stacks for training. Various structures with bright signals that were not part of neurons, like blood vessels, were manually annotated as explicit negative examples. We added an extra penalty to the detection of such structures during training. We generated a training set of 12,000 smaller image tiles from the labeled data. We used the same kinds of data augmentations as for the cell body location. Further details can be found in [62].

**Retrograde tracer segmentations.** Fig 12 illustrates the acquisition and post-processing of retrograde tracer signals. All image sections of the retrograde tracer had 2 color channels, where the first channel (red) contained the cell bodies of the retrograde tracer. The second channel (green) contained the anterograde tracer signal that is also visible in the Tissuecyte microscope (STPT). We utilized the first channel to localize the cell bodies of retrogradely infected neurons and exploited the second channel to align the image to its corresponding Tissuecyte section.

Similar to the detection of cell bodies in the injection sites for anterograde tracers, a U-Net was trained and used for cell body detection. The network took patches (sized 512 Ã–512) as input and was applied to all image sections. Local maxima in the results with a probability larger than 0.5 were considered as detection.

The pipeline registered the second channel with the anterograde tracer signal to the corresponding Tissuecyte section using ANTs. The same image transformation was applied to the first channel and the location of detected cell bodies.

For training the U-Net, about 20,000 patches were randomly sampled from 380 manually annotated image sections (roughly about 20,000 training patches).

**The BMCR STPT template space.** For data integration and investigation, all imaging data was automatically normalized to a volumetric STPT average template with left/right

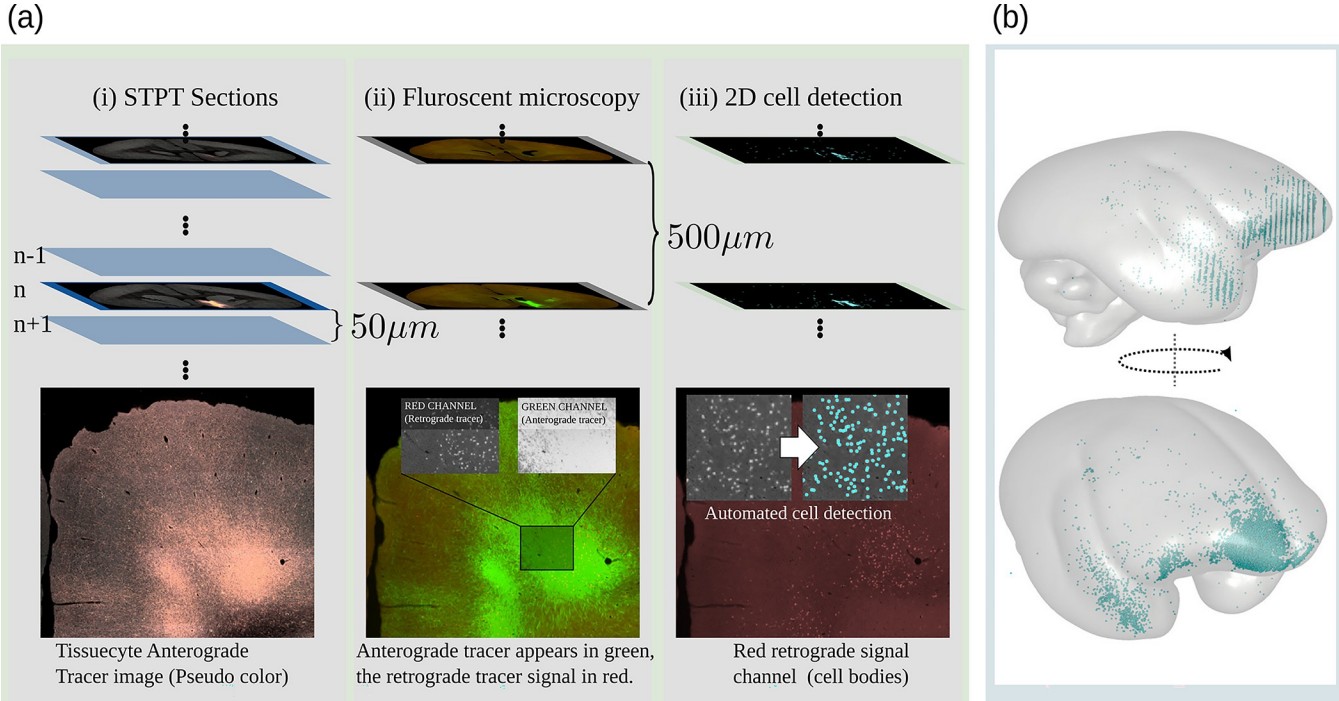

(a)

(b)

(i) STPT Sections

(ii) Fluroscent microscopy

(iii) 2D cell detection

500μm

50μm

n-1
n
n+1

RED CHANNEL (Retrograde tracer)  GREEN CHANNEL (Anterograde tracer)

Automated cell detection

Tissuecyte Anterograde Tracer image (Pseudo color)

Anterograde tracer appears in green, the retrograde tracer signal in red.

Red retrograde signal channel (cell bodies)

**Fig 12. Retrograde tracer segmentations.** (a) Every 10th Tissuecyte section, which corresponds to a 500 μm offset, was recovered and fluorescently immunostained for Cre. Fluorescent images were captured, and a convolutional neural network was applied to detect cell bodies in the image. (b) The cell body locations were mapped to the BMCR template image. BMCR, Brain/MINDS Marmoset Connectivity Resource.

(a)    (b)

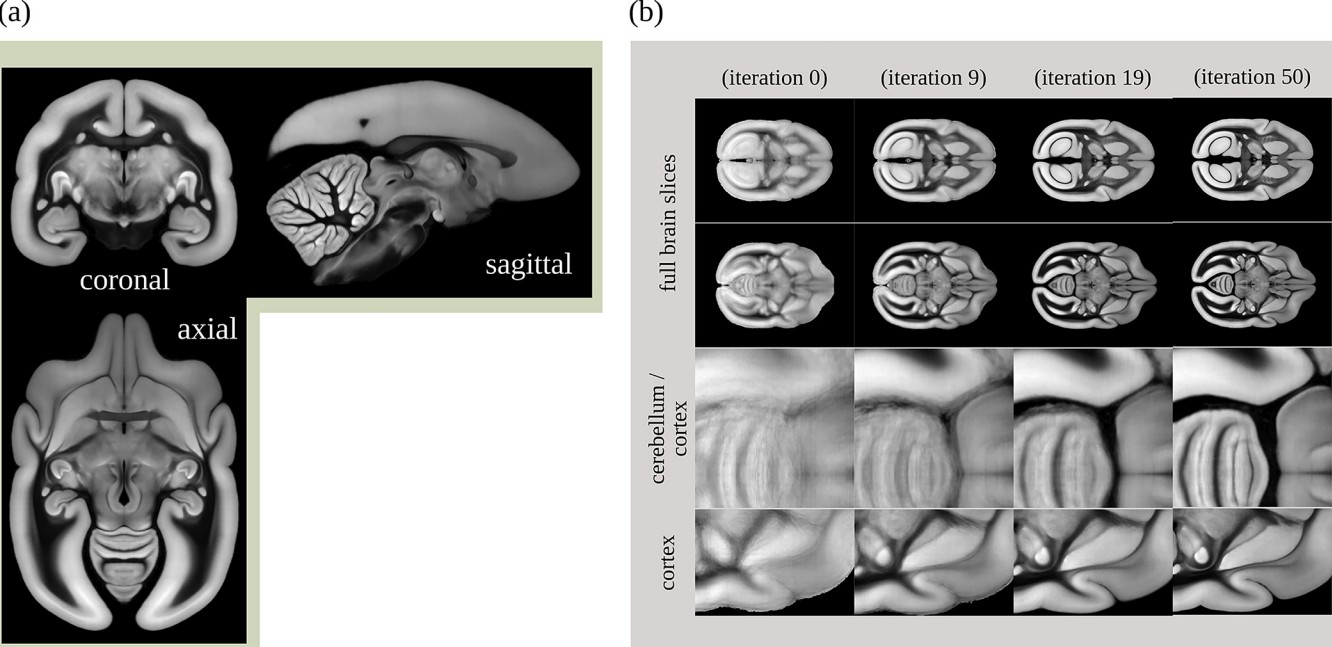

**Fig 13. The BMCR reference image space.** The BMCR reference image space is defined by a population average STPT template image. The STPT template was generated by reiterating the registration of all subjects. Panel (a) shows the STPT template, panel (b) examples the evolution of the template. BMCR, Brain/MINDS Marmoset Connectivity Resource; STPT, serial two-photon tomography.

hemispherical symmetry. The auto-fluorescent background signals (the first channel) of individual STPT images were used for registration and for computing the population average image. Fig 13A shows the template. During averaging, values were inversely weighted by their tracer intensity to suppress image data dominated by neural tracer signals. Areas with missing tissue were excluded as well. The template was generated by a reiterated registration of 36 subjects (including their left/mirrored versions), see Fig 13B. The STPT template has an isotropic resolution of 50 μm.

The spatial resolution of our STPT image sections was sufficiently high to map the microscopy image sections to our template in high resolution. For web deployment, microscopy images were mapped to our template at a high target resolution of $3.0 \times 3.0 \times 50$ μm³. The resulting image stacks contained $9,666 \times 8,166 \times 800$ voxels. All images have been processed and compressed to make them suitable for fast web exploration using either PNG, JPG, or the modern AVIF image file format.

The STPT template was accompanied by a label image annotating the cortex and major subcortical structures such as the thalamus, caudate nucleus, internal capsule, putamen, or hippocampus.

**Atlas mapping.** We computed the transformation fields to map between the BMCR reference space and 3 major marmoset brain atlases which are the following: the Marmoset Brain Mapping atlases version 2 and 3 [19,65], the Marmoset Brain Connectivity Atlas [21,22], and the Brain/MINDS Atlas [20]. In all optimizations, our STPT template was the fixed (target) image. This integration is facilitated by the fact that all current templates adopt the parcellation proposed by [40], ensuring uniformity of histological criteria and nomenclature across studies.

The mapping between the BMCR and Brain/MINDS Atlas was done by computing the warping field between the STPT image template and the T2-weighted population average MRI template (isotropic voxel resolution of 100 μm) of the Brain/MINDS Atlas using ANTs.

The reference image of the Marmoset Brain Connectivity Atlas was a 3D image stack of 63 cortical NISSL stained marmoset brain sections (825×63×550 voxels with a spatial resolution of 0.04×0.5×0.04 μm$^3$). Compared to our STPT template, the sagittal resolution was rather low. To improve the registration, we added a mask for the cortex for both templates as an additional data term for the ANTs optimization (mean square error as metric). The cortex mask for the Marmoset Brain Connectivity Atlas template was generated by fusing all cortical labels in the atlas.

The mapping between the BMCR and the Marmoset Brain Mapping atlas was performed in two steps. The affine registration was done between the STPT template and the symmetric T2-weighted image from the Marmoset Brain Mapping atlas with 80 μm resolution using normalized mutual information as a metric. We added the mean square distance between cortex masks in the SyN step.

Using the warp fields, we mapped the gray matter atlas labels of the Brain/MINDS Atlas, the cortical labels of the Marmoset Brain Connectivity Atlas, and the cortical, subcortical, and white matter labels of the Marmoset Brain Mapping atlas version 2 to the BMCR template space.

**Flatmap stack mapping.** We mapped all 3D tracer image data in the cortex from the BMCR template image to 3D flatmap stacks. A flatmap stack is a 3D image representation of the cortex, where the XY-plane defines the position on the cortex surface and the z-direction defines the relative cortical depth. Flatmap stack mappings are extensions of the flatmaps which are part of the Marmoset Brain Mapping atlas and the Brain/MINDS Atlas. Fig 14 shows an example of cortical anterograde tracer densities mapped to a flatmap stack.

Both the Marmoset Brain Mapping atlas and the Brain/MINDS Atlas share triangulated 3D surfaces that map 3D points of the mid-surface of one hemisphere in the marmoset cortex to a 2D flatmap. The data are publicly available (Marmoset Brain Mapping, https://marmosetbrainmapping.org/atlas.html#v) and Brain/MINDS Atlas, https://dataportal.brainminds.jp/atlas-package-download-main-page/bma-2019-ex-vivo). We utilized these data to map the entire cortex to a 3D image stack, extending the flatmaps with cortical depth.

We first used our warping fields to map the vertices of the 3D surfaces to our STPT template space. Then, we defined the inner border and outer border of the cortex in the STPT template. This step was done manually using the image annotation function in the 3D Slicer tool (https://www.slicer.org/). We computed the normals of the surface for the cortical surface,

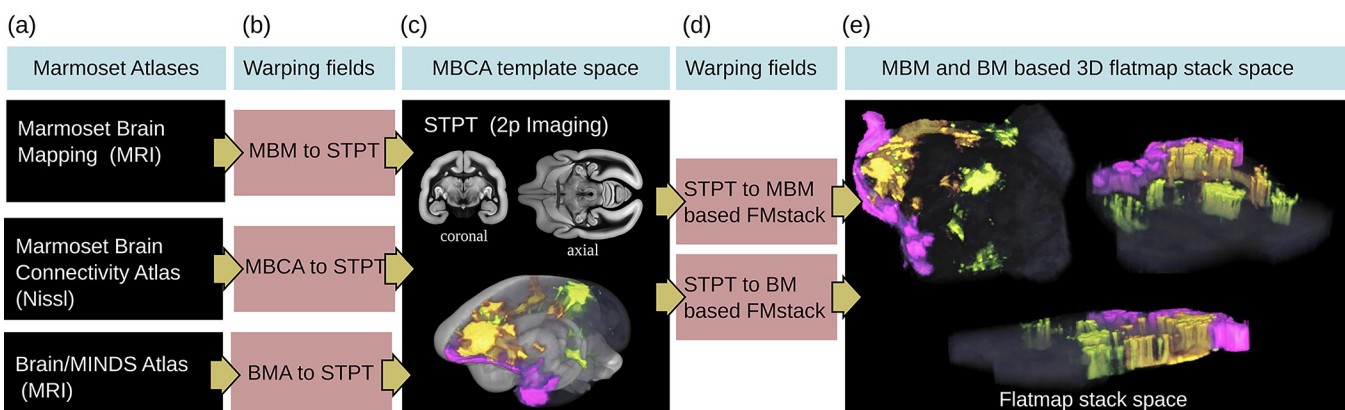

**Fig 14. Mapping between brain image spaces.** The BMCR provides the files for mapping between all major marmoset brain coordinate frameworks. It further can map cortical 3D image data to a flatmap stack using the publicly available ANTs image registration toolkit. This example shows the combined mapping of 3 anterograde tracer images from a 3D brain image to a flatmap stack. BMCR, Brain/MINDS Marmoset Connectivity Resource.

where the normals at the inside pointed towards the cortex, and the normals at the outside pointed away from the cortex. We then used heat propagation to diffuse the directional information within the entire cortex and normalize the result. The directional field defines trajectories that start at the inner cortex boundary, follow the directional field, and terminate at the outer cortex boundary.

We determined the trajectories that intersected with the vertices of the 3D mid-surface of the flatmap data. The 2D counterparts of the intersected flatmap vertices defined the flatmap stack XY coordinates. We traced the trajectories, and the relative position on the trajectory represented the stack depth.

Based on the stack of 3D image coordinates, we generated ANTs warping fields for both the Marmoset Brain Mapping atlas based flatmap stack and the Brain/MINDS Atlas based flatmap stack. The target size of a flatmap stack was 500×500×50 voxels. The cortical atlas data of the Brain/MINDS Atlas, Marmoset Brain Connectivity Atlas, and Marmoset Brain Mapping atlases have been mapped to the flatmap stack image space as well.

**Diffusion MRI data.** The individual HARDI data is also registered to our template space to construct a high-quality structural population average. Registration was done by ANTs using mutual information of the STPT template and T2/b0 contrast. Gradient orientations were re-oriented accordingly. After mapping, the original 128 directions were mapped to left-right symmetric 64 directions using spherical interpolation, and a left-right symmetric population average HARDI was generated. The final resolution of the average HARDI template is 200 μm. Then, new directions obey the symmetry of the STPT image template. Let $S$: $\mathbb{R}^3 \times S_2 \rightarrow \mathbb{R}_+$ be the averaged HARDI image, whereas $S_2$ is the unit sphere. The new directions obeys the angular and spatial symmetries $S(x,y,z,n_x,n_y,n_z) = S(-x,y,z,-n_x,n_y,n_z)$ and $S(\mathbf{x},\mathbf{n}) = S(\mathbf{x},-\mathbf{n})$. We used our in-house optimization algorithm to place charged particles that repel each other on a sphere until they reach an equilibrium state with equal distances between each pair based on the previous work of [66]. The code can be found in our BMCR code repository (filename: HARDI_sym.m).

For the HARDI template, we added both the registered HARDI images and their mirrored counterparts to increase the number of samples and to ensure symmetry with respect to the x axis. Several images suffered minor tissue damage due to the extraction of the brain. We excluded all images with major imaging artifacts or tissue damage. In order to suppress the effect of minor tissue damages in the remaining 23 images, we used a weighted average to compute the average HARDI template, similar to the STPT template computation.

From the population average HARDI template, we generated standard diffusion metrics such as diffusivities and fractional anisotropy. We further applied global tractography [67]. We build a symmetric tractogram by mirroring the streamlines with respect to the left and right hemisphere. For each dataset, we took the injection site as mask, and selected all touching streamlines using the mrtrix3 tckedit algorithm [68]. From the streamlines, we generated streamline density images [69] for each injection site. Fig 15 outlines the generation of streamline density images.

**Nissl and backlit.** After the STPT imaging, the sections were imaged twice. Once before (backlit) and after Nissl staining. For backlit imaging, which reveals features of the brain myelination, the slices were collected from the STPT microscope and mounted onto slides. Then after imaging, stained for Nissl bodies and imaged a second time. In both steps, physical deformations happened due to the mounting, staining, or decaying processes. The major deformations occurred during the initial mounting process. We used a multimodal image registration to undo the deformations in the images. Fig 16 outlines the registration pipeline. First, the Nissl image was mapped back to the backlit image. Then, the backlit image was mapped back to its corresponding STPT image slice. Applying the concatenated warp fields mapped both

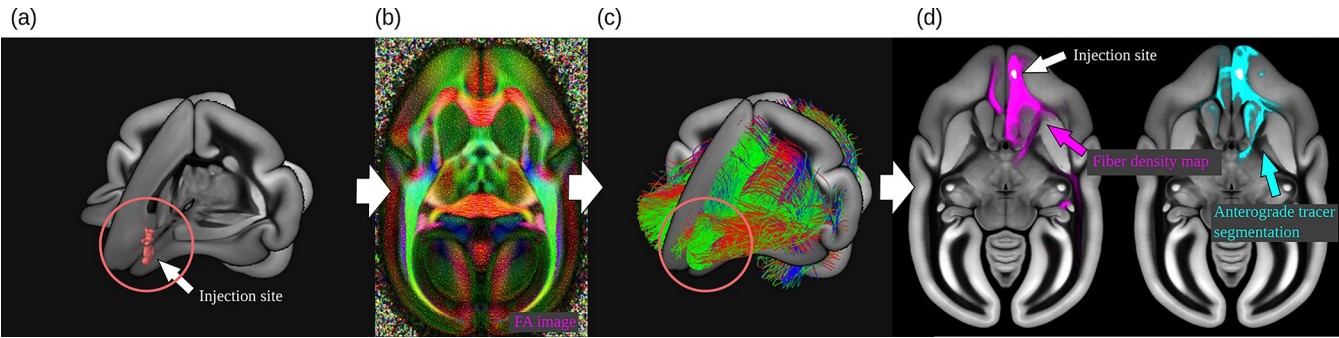

**Fig 15. Generation of streamline density images.** (a) Based on an injection site location, (b, c) the pipeline selects all intersecting streamlines from a global streamline pool (a large tractogram). (d) Shows streamline density and tracer density, both associated with the same injection site.

images back to the image space of the original STPT image section. In an initial trial, the registration between the 3 different image modalities occasionally failed due to the major visual differences that hindered automation. We employed a semi-supervised image-to-image translation technique that adjusted the contrast of Nissl and backlit images to resemble the STPT template image during the initial affine registration, resolving the issue. The image-to-image translation was realized with a generative adversarial network. Further details and code are publicly available online [70].

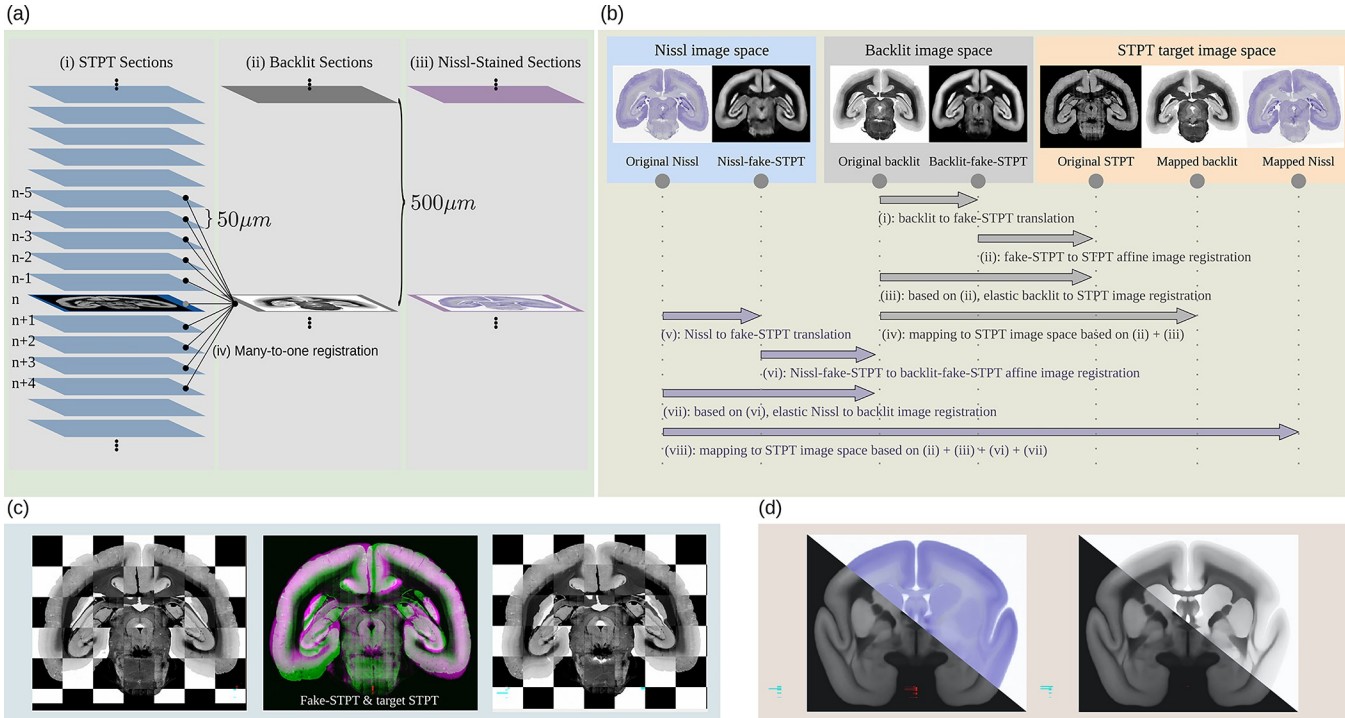

**Fig 16. The Nissl and backlit registration pipeline.** (a) For every 10th STPT tissue section, we take an additional backlit and a Nissl image with a second microscope. (b) Each time a section is moved or stained, it undergoes physical deformations. We established a robust image registration pipeline that reliably aligns the Backlit/Nissl images with the STPT images in a fully automated manner. (c) An example of a backlit image section before and after alignment. (d) After mapping the Nissl/Backlit images of our entire population to the STPT reference template, we computed population averages. STPT, serial two-photon tomography.

## Integration of the Marmoset Brain Connectivity Atlas

We used the Marmoset Brain Connectivity Atlas API (https://github.com/Neuroinflab/analysis.marmosetbrain.org/wiki/Application-Programming-Interface) [21,22] to download the retrograde cell datasets. We first got a list of all available datasets via the "injections" command. Then, for each injection, we downloaded the list of cell locations via the "cells?injection_id =" command. The cell locations were given within the Paxinos stereotaxic reference space [40]. Cells were labeled as "supragranular" and "infragranular" based on their cortical location with respect to cortical layer IV.

We used our ANTs transformation field to map the cell positions and injection site locations to the BMCR. For all injections, we created cell density images. We generated images with an isotropic spatial resolution of 100 μm and images with 400 μm resolution. Similar to the data that is publicly available on the Marmoset Brain Connectivity Atlas web portal, we generated 3 types of cell density maps. Each one for "infragranular," "supragranular," and for combination of both cell categories. In addition, all density maps have been mapped to flatmap stacks using our ANTs transformation field.

## Pipeline program code

The source code for the pipeline and the code for generating the flatmap stack warping fields are publicly available from https://doi.org/10.5281/zenodo.7906530 and from https://doi.org/10.5281/zenodo.7906607, respectively.

Several pieces of code that orchestrate workload distribution are tailored to our SLURM HPC cluster. However, each individual processing script focuses on a single pipeline step and can be executed independently. Experts should be able to tailor them to their needs. Table 5 lists the most important scripts.

## Data validation

All data were visually screened by an expert using the BMCR-Explorer and the Nora-StackApp visualization tools. The screening did not reveal noteworthy misaligned parts in the images.

**Table 5. Major processing scripts used in the pipeline.**

| Script name | Description |
| --- | --- |
| kn_pipeline_meso_get_t2n.py | Converts 2D STPT sections to Nifti stacks |
| kn_pipeline_meso_reg2TCstd_org.py | Register STPT image stacks to STPT template |
| kn_pipeline_meso_inj_seg.py | Rough localization of the injection site |
| kn_pipeline_meso_NN_cells.py | Single-cell detection in the injection site using a DNN |
| kn_pipeline_meso_NN_cells_3D.py | Computing cell density images for the injection site |
| kn_pipeline_meso_NN_tracerseg.py | DNN-based anterograde tracer segmentation |
| kn_pipeline_meso_NN_tracerseg_3D.py | 3D image stack generation |
| kn_pipeline_meso_norm_tracer.py | Anterograde tracer normalization |
| kn_pipeline_meso_NISSL_2D.py | Registering the Nifti images and creating stacks |
| kn_pipeline_meso_apply_trafos_TC.py | Moving all images to the STPT/BMA/MBMv2+3 image spaces |
| kn_pipeline_meso_map_pts2std.py | Moving point data to the STPT image space |
| kn_pipeline_meso_flatmap.py | Mapping data to the BMA and MBM flatmaps |
| kn_pipeline_meso_flatmap_stack.py | Mapping to flatmap stacks |
| kn_pipeline_meso_DWI.py | Creating streamline density maps for the injection site |
| kn_pipeline_meso_map_highres.py | Mapping image data to high-resolution template space |

STPT, serial two-photon tomography.

We did not find any considerable problems with the automated detection of the injection sites, the segmentation of the anterograde tracer signal, nor the localization of the retrograde tracer signal.

The age of marmosets ranges from 2.5 years to about 10 years. Tables 1 and 2 provide details. We did not detect significant correlations between age and tracer intensity (Pearson correlation $p < 0.8304$), nor between age and tracer signal volume ($p < 0.4716$) or injection site volume ($p < 0.3927$), see Table 6. For all cases, the 0 hypothesis was tested by a $n = 10k$ permutation test. On the other hand, the volume of the injection site seems to correlate with the volume of the detected tracer signal ($p < 0.0001$), which can be attributed to the increasing number of neurons loaded with fluorescent proteins.

In addition to visual image validation, we performed quantitative validations as described below.

**STPT image registration.** To validate the image registration, we defined 20 landmarks within the marmoset brain. Sixteen of them were pairs of landmarks that exist in both the left and the right brain hemispheres. The remaining 4 were located on the brain mid-surface. Fig 17 shows the landmarks in detail. The landmarks were defined in the STPT template space, and then mapped to every other marmoset brain (we used 26 out of 52 images for validation, see Tables 2 and 3 for a list of the marmosets).

The mapping of landmarks was done automatically and was additionally performed manually 3 times by 3 different experts (26 images × 20 landmarks × 4). We utilized the Nora imaging platform (https://www.nora-imaging.com/) to share images with the 3 experts. We used the Marker Tools of Nora for adjusting landmark locations. The experts could explore the landmarks in the STPT template and had to determine the positions of their corresponding counterparts within the 26 images. We performed the same procedure for 4 external marmoset brain atlases. We share all landmarks (automatic and manual) as json files.

To compare manual with automated mapping, we compared 2 groups as shown in Fig 18E. The first group represented the displacements between the same landmarks placed by 3 different human experts. The second group the displacement between a landmark placed by an expert and the corresponding automatically mapped landmark. For each group, we chose the median displacement for each landmark resulting in 26 × 20 landmark displacements per

**Table 6. Anterograde tracer data correlation analysis.**

|  | Data | Pearson | *P*-value | Spearman | *P*-value |
|---|---|---|---|---|---|
| 1 | age vs. med(c2) | −0.06 | <0.8304 | −0.03 | <0.6867 |
| 2 | age vs. sum(c2) | −0.12 | <0.3499 | −0.13 | <0.4084 |
| 3 | age vs. sum(td) | −0.15 | <0.4716 | −0.10 | <0.2888 |
| 4 | age vs. sum(inj) | 0.08 | <0.3620 | 0.13 | <0.5648 |
| 5 | age vs. vol(inj) | 0.14 | <0.3927 | 0.12 | <0.3180 |
| 6 | sum(inj) vs. sum(c2) | 0.42 | <0.0012 | 0.46 | <0.0028 |
| 7 | sum(inj) vs. med(c2) | −0.10 | <0.8407 | −0.03 | <0.4833 |
| 8 | sum(inj) vs. sum(td) | 0.70 | <0.0001 | 0.72 | <0.0001 |
| 9 | vol(inj) vs. sum(c2) | 0.27 | <0.0224 | 0.31 | <0.0561 |
| 10 | vol(inj) vs. med(c2) | −0.20 | <0.3437 | −0.13 | <0.1468 |
| 11 | vol(inj) vs. sum(td) | 0.62 | <0.0001 | 0.64 | <0.0001 |

Correlations between anterograde tracer signal attributes, injection site attributes and age. The table lists the Pearson and Spearman correlation coefficients together with the *P*-value of the 0-hypotheses based on a 10k permutation test. Attributes that seem to be correlated are shown in red. The term med(c2) means median of the anterograde tracer signal intensity, sum(c2) the sum of the tracer intensity, sum(td) the sum of tracer positive voxels, sum(inj) the estimated number of cellsin the injection site, and vol(inj) the volume of the injection site.

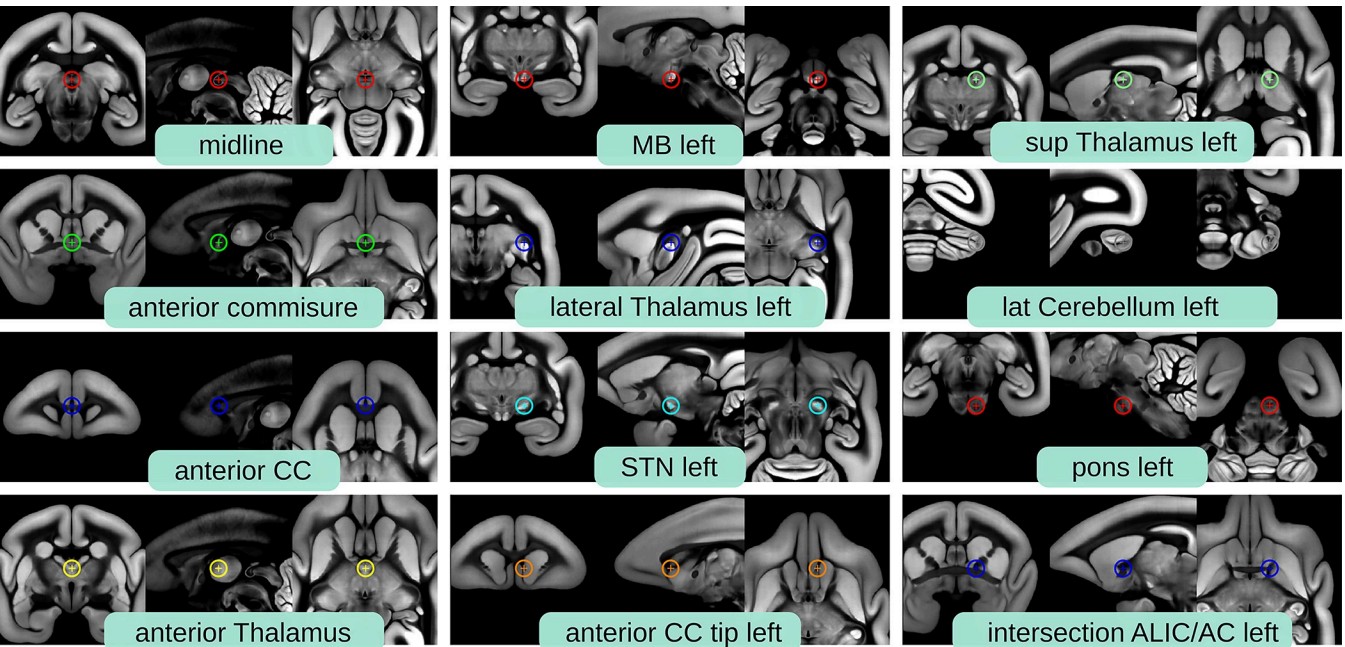

**Fig 17. Validation landmarks.** The image shows the 20 landmarks that we used for validating the image registration (for landmarks that appear in both brain hemispheres, only the left landmark is shown). We provide public access to the landmark locations. The data can be downloaded from our data repository (https://doi.org/10.60178/cbs.20230630-001, filename: validation_landmarks_atlases.zip).

group. Fig 18A, which depicts the mean displacement sorted in increasing order, shows that automated landmark positioning maintains the performance of human annotations. We further quantified how often automation was in a better agreement with human annotation than the agreement between 2 different humans. Fig 18B shows the results. Again, automation performs remarkably well, except for the midline and superior thalamus landmarks. It is worth noting that the agreement for these landmarks was high for both automation/manual and manual/manual pairs, as shown in Fig 18B.

**Atlas image registration.** We repeated the same experiment, where this time we mapped the landmarks to the MRI template of the Brain/MINDS Atlas, the MRI templates of the version 2 and version 3 atlases of the Marmoset Brain Mapping project (Marmoset Brain Mapping v2 and v3), and the Nissl template of the Marmoset Brain Connectivity Atlas. For the latter, we only considered the landmarks on the mid-surface and the left brain hemisphere, since the atlas covers only 1 brain hemisphere. Fig 18C shows the displacement in the increasing order and Fig 18D a direct comparison between the 2 groups. In most cases, the automation showed better agreement with the human annotation than the annotations within the group of human experts. Only for the landmarks in the lateral cerebellum and lateral thalamus were the agreement between the human annotations more frequent than between the automation and the humans. But even here, the shift was rather moderate in both cases compared to the other landmarks.

Since all atlases provide cortical annotations, we used these annotations to validate the cortical overlap after image registration. Fig 18F shows the overlap of cortical labels from an atlas and the STPT template after image registration together with their Dice score (normalized intersection over union $Dice = \frac{2|mask_1 \cap mask_2|}{|mask_1| + |mask_2|}$). For all atlases, the agreement is remarkably high (Marmoset Brain Connectivity Atlas 96.1%, Marmoset Brain Mapping atlas v2 95.9%,

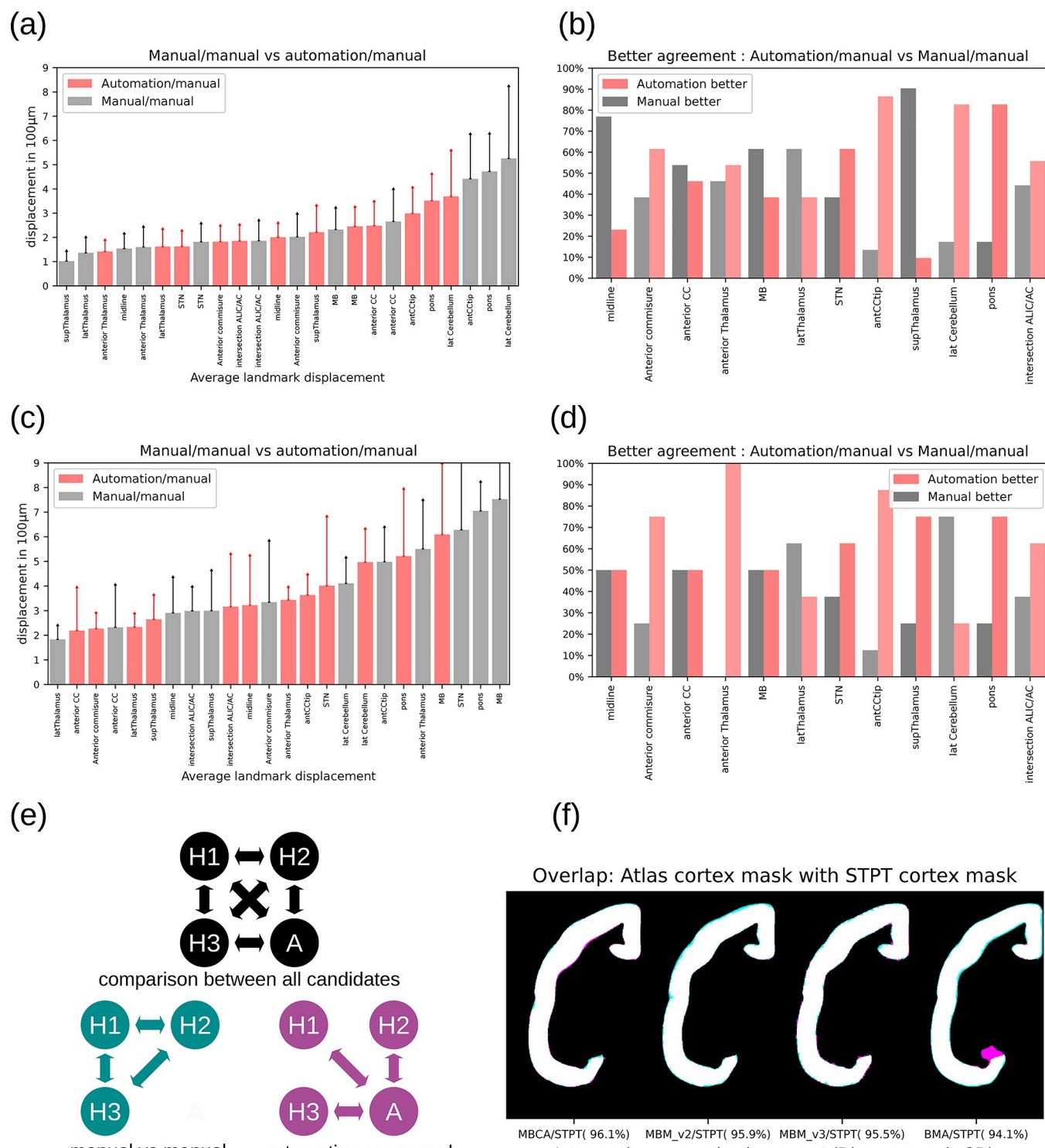

**Fig 18. Evaluation of the image registration.** Quantitative evaluation of the image registration based on landmarks and cortical atlas labels shows that the pipeline maintains or even exceeds manual accuracy. Panel (a) shows the average displacement of landmarks placed by different human experts (manual/manual) compared to the displacement of landmarks between automation and a human expert (automation/manual). The plot in panel (b) shows how often an automated landmark position was closer to a manually placed landmark location than the locations from 2 different human annotators. Panels (c) and (d) show the results for the 4 other marmoset brain atlases and our STPT template space. The procedures were the same as for panels (a) and (b). Panel (e) We split the annotations in 2 groups. We compared the median agreement of landmark locations among humans (manual vs. manual) with the median agreement among human and automation. Panel (f) shows the overlap of the cortex annotation from the atlases with our own annotation after image registration (MBCA:

Marmoset Brain Connectivity Atlas, MBM: Marmoset Brain Mapping, BMA: Brain/MINDS Atlas). The cortex mask of the STPT is drawn in cyan, the masks of the other templates in magenta. Overlaps appear in white. Data availability: The source code that generated the plot and images is publicly available (https://doi.org/10.5281/zenodo.7906530, filenames: BMCR_Fig 18_ab.ipynb and BMCR_Fig 18_cdf.ipynb), and the corresponding data is publicly available on the CBS Data Sharing Platform (https://doi.org/10.60178/cbs.20230630-001). STPT, serial two-photon tomography.

Marmoset Brain Mapping atlas v3 95.4%, and Brain/MINDS Atlas 94.1%) meaning that cortical boundaries have been registered precisely.

**Injection site.** The injection site can be spotted quite easily as the brightest area in the 3D reconstruction of the image of the third STPT channel. The infected cell bodies appear significantly brighter than the background, and the signal drops rapidly towards zero outside the injection site. The pipeline detects the tracer injection site in 2 steps: coarse localization followed by a cell body detection.

To quantitatively validate the coarse localization, we asked 3 experts to manually mask the injection site in every other dataset (alphabetic order, 26 of 56 datasets) and we compared the overlap with the automatic estimation. We utilized the Nora imaging platform (https://www.nora-imaging.com/) to share 3D image stacks of the third STPT channel with the experts. When opening the images, the view port was automatically centered with respect to the brightest spot in the volume that corresponded with the injection sites in all cases. Each expert manually created a binary mask covering the injection site. We used the masks to validate the automated rough localization of the injection site. The masks are shared as 3D Nifti image stacks as part of the BMCR dataset.

A median overlap of 70% and a minimum overlap of no less than 34% showed that the automatic localization found the correct brain region in all cases. To validate details, we measured the cell recognition accuracy of our cell detection deep neural network. We manually labeled 3,524 cells in 5 different image slices from 5 different injection sites that were not part of our training set. Quantitative analysis revealed that the cell detection network detected 91% of manually labeled cells with an accuracy of 85%, showing high precision and accuracy. Further details and an additional experiment demonstrating that our deep learning approach outperforms a standard heuristic cell body detection approach can be found in our technical report [62].

**Retrograde tracer detection.** We assessed the automatic detection of retrograde cell bodies using manually labeled data. A human expert manually labeled the location of retrograde cell bodies in 455 slices from 11 of the 19 subjects with retrograde tracer data. To select the best-performing model for the U-Net, we performed 4-fold cross-validation during training. Data from 380 slices were used for training, and the remaining data were used for validation. Table 7 shows the results for the entire labeled dataset (TP = true positives, FP = false positives, FN = false negatives, PRE = precision, SEN = sensitivity (recall)). We predicted more cells than indicated in the manual labels. Visual inspection of the results indicated that the algorithm performed better overall compared to manual annotation. The automation was often able to detect single cells in dense cell clusters that were difficult to process manually due to the large amount of cell bodies. The algorithm also found small clusters of cells in sections that could easily be missed by manual annotation. For cross-validation, we used the marmosets with ids R01_0088, R01_0090, R01_0091, R01_0092, R01_0094, R01_0095, R01_0098, R01_0103, R01_0112, R01_0114, and R04_0095.

**Table 7. Retrograde cell body detection.**

| Labels | Prediction | TP rate | FP rate | FN rate | PRE | SEN |
|---|---|---|---|---|---|---|
| 138,096 | 160,624 | 0.9442 | 0.1900 | 0.0485 | 0.8100 | 0.9511 |

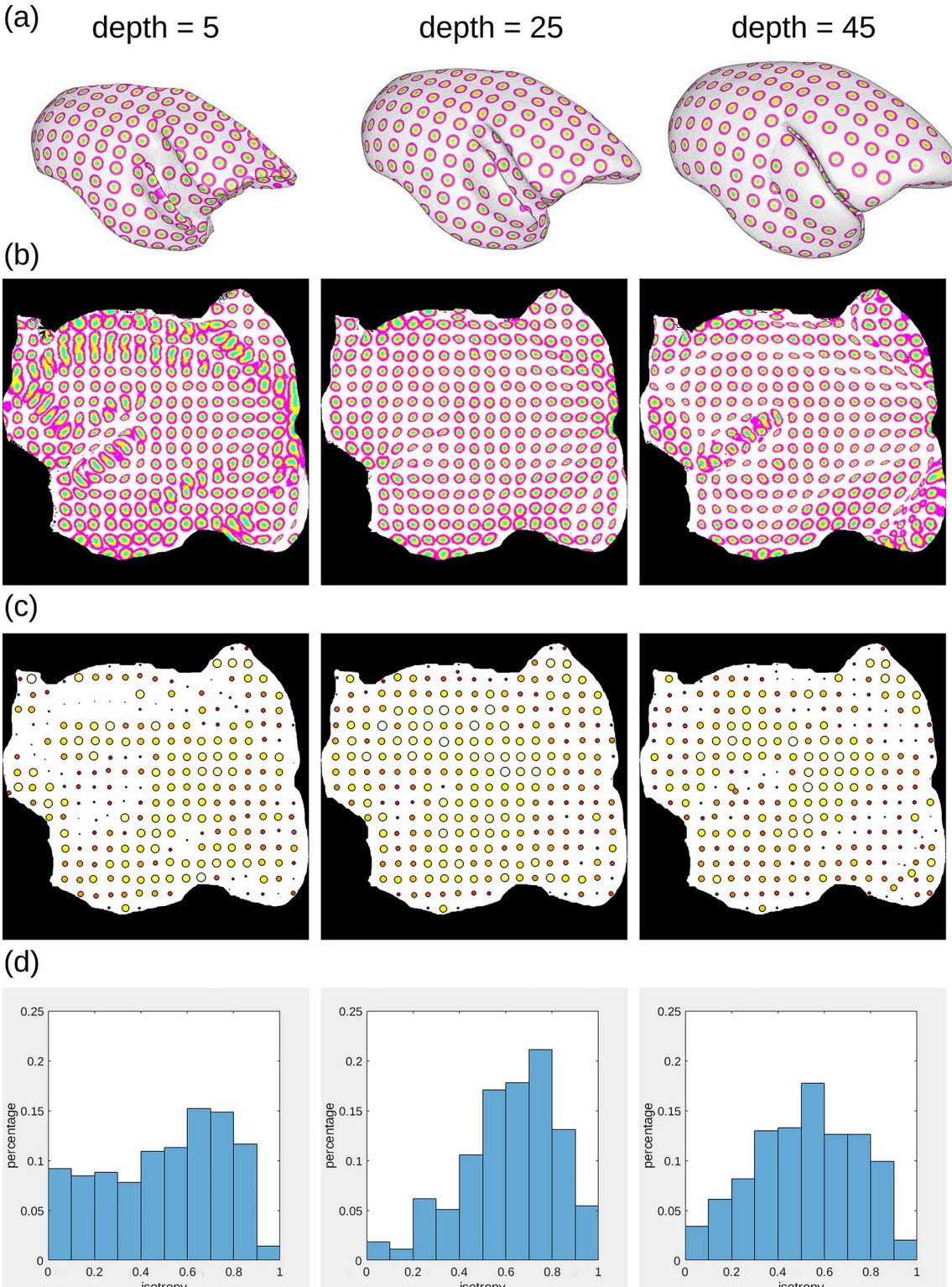

**Fig 19. Evaluating the flatmap stack mapping.** We qualitatively and quantitatively validated the deformations of the flatmap stack mapping for 3 different levels of cortical depth. Each column in this figure corresponds to a level. Level 5 is located close to the WM/ GM border, while level 45 is close to the pia-matter. Anisotropic deformations appear in parts with gyrifications in either the lower or upper levels. Panel (a) shows the intersection with spheres (Volume rendering), panel (b) images of the spheres mapped back to the corresponding flatmap stack layer. Row (c) shows the degree of isotropy as heat maps and circle size, and (d) histograms of the degrees

of isotropy. Data availability: The source code that generated the plots and images is publicly available (https://doi.org/10.5281/zenodo.7906530, filename: BMCR_Fig 19.m), and the corresponding data is publicly available on the CBS Data Sharing Platform (https://doi.org/10.60178/cbs.20230630-001).

**Flatmap stack mapping.** To validate deformations of the flatmap stack mapping, we evaluated deformations for 3 cortical depth levels. The flatmap stack equidistantly divides the cortex in 50 different layers. We validated the flatmap stack layers 5, 25, and 45 using the Brain/MINDS Atlas based flatmap stack. We placed seed points equidistantly into a flatmap stack layer, mapped the points back into the STPT reference space, and drew isotropic 3D spheres around them. Then, we mapped them back into the flatmap stack to visualize their deformations. Fig 19 shows the results. Fig 19A shows the intersection of the corresponding cortical layers with the spheres, Fig 19B their back-projections. An ideal mapping would map the spheres back to circles with unified radii. That is impractical to achieve because of the shape and topology of the cortex. Each back-projected sphere defines an ellipsoid for which we computed its principal components. Based on that, we calculated the level of isotropy for each ellipsoid ranging from 0 to 1. Fig 19C illustrates the isotropy as heat map, Fig 19D as histograms. The results show that the flatmap stack maintains a fairly equal size of the spheres for most of the cortical surface area. However, anisotropic deformations appear in parts with gyrifications or strong curvature, which should be taken into account when working with the images.

## Ethics statement

All experimental procedures were carried out following the National Institute of Health Guide for the Care and Use of Laboratory Animals (NIH Publications No. 80–23) revised in 1996 and the Japanese Physiological Society's Guiding Principles for the Care and Use of Animals in the Field of Physiological Science and were approved by the Experimental Animal Committee of RIKEN (W2020-2-009(2)).

## Data access

The **BMCR-Explorer** is publicly available at the following link: http://bmca.riken.jp/. The RIKEN CBS data repository provides access to the main resources, such as Nifti image stacks (refer to Table 1) and auxiliary data like warping fields (refer to Table 4), as well as the **Nora-StackApp** at https://doi.org/10.60178/cbs.20230630-001.

Source code for generating Figs 5, 7, 8, 18, and 19 is publicly available from https://doi.org/10.5281/zenodo.7906530 ("BMCR_Figures" subfolder).

You can also find all the links to access the BMCR-Explorer and download data and tools on the Brain/MINDS data portal: https://dataportal.brainminds.jp/marmoset-connectivity-atlas.

## Author Contributions

**Conceptualization:** Henrik Skibbe, Ken Nakae, Kenji Doya, Tetsuo Yamamori, Shin Ishii, Akiya Watakabe.

**Data curation:** Junichi Hata, Tetsuo Yamamori, Akiya Watakabe.

**Formal analysis:** Henrik Skibbe.

**Methodology:** Henrik Skibbe, Muhammad Febrian Rachmadi, Ken Nakae, Carlos Enrique Gutierrez, Marco Reisert, Akiya Watakabe.

**Resources:** Henrik Skibbe, Junichi Hata, Kenji Doya, Piotr Majka, Hideyuki Okano, Tetsuo Yamamori, Shin Ishii.

**Software:** Henrik Skibbe, Muhammad Febrian Rachmadi, Marco Reisert.

**Supervision:** Henrik Skibbe.

**Validation:** Henrik Skibbe, Ken Nakae, Carlos Enrique Gutierrez, Hiromichi Tsukada, Matthias Schlachter, Akiya Watakabe.

**Visualization:** Henrik Skibbe.

**Writing – original draft:** Henrik Skibbe, Marco Reisert, Akiya Watakabe.

**Writing – review & editing:** Henrik Skibbe, Ken Nakae, Carlos Enrique Gutierrez, Charissa Poon, Piotr Majka, Marcello G. P. Rosa, Tetsuo Yamamori, Marco Reisert, Akiya Watakabe.

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
