## [Editor Report · Decision Letter 0]

19 Jan 2023

Dear Dr Skibbe, 

Thank you for submitting your manuscript entitled "The Brain/MINDS Marmoset Connectivity Resource: an open access multimodal platform integrating cellular-level bidirectional tracing and tractography in the primate brain" for consideration as a Methods and Resources by PLOS Biology.

Your manuscript has now been evaluated by the PLOS Biology editorial staff, as well as by two academic editors with relevant expertise, and I am writing to let you know that we would like to send your submission out for external peer review. Please note that, given some of the comments from our Academic Editors regarding the extent of new data versus merging of data from the prior marmoset databases, we will be looking for strong reviewer enthusiasm on overall advance grounds for this Methods and Resources submission.

Before we can send your manuscript to reviewers, we need you to complete your submission by providing the metadata that is required for full assessment. To this end, please login to Editorial Manager where you will find the paper in the 'Submissions Needing Revisions' folder on your homepage. Please click 'Revise Submission' from the Action Links and complete all additional questions in the submission questionnaire. During this process, you will be offered the option to suggest reviewers with appropriate expertise to evaluate your work. We do find such suggestions helpful if you are so inclined to provide them. You will also be offered the option to exclude up to 3 individuals.

Once your full submission is complete, your paper will undergo a series of checks in preparation for peer review. After your manuscript has passed the checks it will be sent out for review. To provide the metadata for your submission, please Login to Editorial Manager (https://www.editorialmanager.com/pbiology) within two working days, i.e. by Jan 21 2023 11:59PM.

Kind regards,

Kris

Kris Dickson, Ph.D., (she/her)

Neurosciences Senior Editor/Section Manager

PLOS Biology

kdickson@plos.org

---

## [Decision Letter · Decision Letter 1]

16 Mar 2023

Dear Dr Skibbe,

Thank you for your patience while your manuscript "The Brain/MINDS Marmoset Connectivity Resource: an open access multimodal platform integrating cellular-level bidirectional tracing and tractography in the primate brain" went through peer-review at PLOS Biology. Please accept my apologies for the delays that you have experienced during the peer review process. I am now handling your manuscript since Kris Dickson has now moved on from PLOS Biology. Your manuscript has now been evaluated by the PLOS Biology editors, an Academic Editor with relevant expertise, and by three independent reviewers.

The reviews are pasted below. As you will see, they are all very positive about your Resource and think it is of high quality. In light of the reviews, we are pleased to offer you the opportunity to address the comments from the reviewers in a revision that we anticipate should not take you very long. Specifically, the reviewers note that the organization and presentation of the manuscript should be made clearer to allow the reader to follow the methodology, as well as raising concerns that they unable to use Nora StackApp or download the tracer data from the website. We will then assess your revised manuscript and your response to the reviewers' comments with our Academic Editor aiming to avoid further rounds of peer-review, although might need to consult with the reviewers, depending on the nature of the revisions.

In addition, I would also be grateful if you could please address the following data and other policy-related requests that I have listed below (A-E):

(A)We would like to suggest that the following modification the title, to make it more accessible for our broad readership:

"The Brain/MINDS Marmoset Connectivity Resource: an open access platform for cellular-level tracing and tractography in the primate brain"

(B) In the Methods section of the manuscript, please provide additional details regarding housing conditions, feeding regimens, environmental enrichment, and all relevant steps taken to alleviate suffering of the animals (anesthesia, analgesia, details about humane endpoints, euthanasia, etc.). Also indicate how often animal care staff monitored the health and well-being of the animals and the criteria used to make such assessments. Lastly, specify the disposition of animals at the end of the study (e.g. euthanasia, returned to home colony, etc.). If animals were euthanized following the study, please provide the method of sacrifice.

(C) You may be aware of the PLOS Data Policy, which requires that all data be made available without restriction: http://journals.plos.org/plosbiology/s/data-availability. For more information, please also see this editorial: http://dx.doi.org/10.1371/journal.pbio.1001797

-Supplementary files (e.g., excel). Please ensure that all data files are uploaded as 'Supporting Information' and are invariably referred to (in the manuscript, figure legends, and the Description field when uploading your files) using the following format verbatim: S1 Data, S2 Data, etc. Multiple panels of a single or even several figures can be included as multiple sheets in one excel file that is saved using exactly the following convention: S1_Data.xlsx (using an underscore).

- Deposition in a publicly available repository. Please also provide the accession code or a reviewer link so that we may view your data before publication.

Figure 7A-C, 18A-D, 19D

(D) Please also ensure that each of the relevant figure legends in your manuscript include information on *WHERE THE UNDERLYING DATA CAN BE FOUND*, and ensure your supplemental data file/s has a legend.

(E) In line with the reviewer comments, please ensure that the tracer data is publicly available from the Brain/MINDS data portal and that the Nora StackApp is able to used by the user.

We expect to receive your revised manuscript within 2 months. Please email us (plosbiology@plos.org) if you have any questions or concerns, or would like to request an extension. 

**IMPORTANT - SUBMITTING YOUR REVISION**

*Resubmission Checklist*

*Published Peer Review*

*PLOS Data Policy*

*Blot and Gel Data Policy*

Sincerely,

Richard

Richard Hodge, PhD

Associate Editor, PLOS Biology

rhodge@plos.org

REVIEWS:

Reviewer #1: The authors present a multi-modal resource platform of the marmoset brain expected to be of broad interest in the community. Especially, the integration of both anterograde and retrograde tracers that have been reconstructed in 3D image volumes and the collection of ex vivo diffusion MRI on the same brain. The authors have constructed a 3D template of their multi-modal data and warped existing marmoset databases to their template increasing the data material and generality of usage across databases. Besides a comprehensive data processing pipeline the authors also demonstrate a visualization tool. The data material and tool presented look highly convincing and of high quality. That said the manuscript could be more logically organized for the reader to follow such a complex setup and lacks comprehensive critical proofreading - too many sentences are too hard to follow. The authors have selected to give "validation" examples of diffusion MRI-based tractography versus tracer's projections to demonstrate the usage of multimodality. It is basically a good idea and the authors do have unique data material including both anterograde and retrograde tracers. However, the examples used appear sloppy performed/unclearly presented. How are their validation findings related to the state of art validation and tractography issues? Basically, the diffusion MRI modality data part needs to be tightened up. In the methods section experimental details are missing. How was the brain prepared for ex vivo MRI, how was it prepared in the MRI scanner, temperature drift of tissue that can impact diffusivity, argue why a b-value of 3000 s/mm^2 was chosen, how are the directions organized on the unit sphere, how long was the scan time, etc.? The ex vivo setup lacks references to relevant literature. Further, the terminology used for tractography is confusing and can be misunderstood when compared with tracer data. Suggest avoiding using "Fiber" in terms of tractography but using "streamlines" and "streamline density" instead of "fiber density". Fiber density can be misunderstood as related to axon density, but the number of streamlines is not related to axon density but more to how many streamlines emanate from a seeding region. Generally, the method sections could be clearer. The discussion includes "Originality and significance" but not clear why it is needed. Overall, maybe the authors should consider simplifying the manuscript so the reader can appreciate the interesting multi-modal resource they wish to provide. 

Reviewer #2 (Stefan Everling, signs review): Skibbe and colleagues present here an open access platform for anterograde and some retrograde tracer data for different prefrontal areas in the marmoset. This is a fantastic resource! I have already played around with the BMCR viewer for hours and this dataset will be an extremely valuable resource for anyone working in marmoset neuroscience or who is interested in aspects comparative neuranatomy. I really would like to thank the authors for collecting these data and for putting this very impressive resource together. I cannot comment on the fine details of the actual processing pipeline but the results look amazing and seem to be of very high quality. I only have a few comments. 

Major comments

1. The BMCR explorer is easy and pretty intuitive to use. The functionality is truely impressive! However, I could not get the NoraStack app to work at all and therefore could not evaluate it.

2. I also could not download the actual tracer data from the website. Are they actually now open access? I think they should be so that researchers can analyze them with their own analysis tools. The current download folder is password protected and none of the supplied passwords worked. 

3. The authors should really include the (lower resolution) tracer data as nifti files so that they can easily be combined with other imaging data such as MRI and fMRI by other groups. Basically similar to what is already included in marmoset brain connectivity atlas (marmosetbrain.org). The BMCR viewer is really nice but having the data also in nifti format would be extremely useful.

Minor comments

P3,line 26-27: the authors could also mention the granular prefrontal cortex of primates here

P11, line 197 "generated"

P16, line 367: replace "was" with "of"

P17, line 399: "lunch" should be "launch"

Lines 503-505, figure: change to "sagittal"

Reviewer #3 (Matthew F. Glasser, signs review): The authors present what is a super impressive body of work on invasive and non-invasive marmoset connectivity. I found the software somewhat more frustrating (had some success with the BMCR-Explorer but could not get the Noraview to work on Mac with multiple error messages when attempting to load files or folders). I saw that both software had a concept of a "scene" where one saves the state of the software. That could have been used to better advantage to hold the hand of the novice user, as the readme for Noraview was extremely terse. My main suggestion would be to get some colleagues unfamiliar with the tools to try them out and give notes on what works well and what doesn't. It could be a little clearer that the diffusion data were from multiple animals (in some places this is clear, others less so). It was unclear to me why the diffusion data and templates in general needed to be symmetrized. Perhaps this could be justified.

---

## [Editor Report · Decision Letter 2]

11 May 2023

Dear Henrik,

On behalf of my colleagues and the Academic Editor, Henry Kennedy, I am pleased to say that we can in principle accept your manuscript for publication, provided you address any remaining formatting and reporting issues. These will be detailed in an email you should receive within 2-3 business days from our colleagues in the journal operations team; no action is required from you until then. Please note that we will not be able to formally accept your manuscript and schedule it for publication until you have completed any requested changes.

PRESS

Kind regards, 

Richard

Richard Hodge, PhD

Associate Editor, PLOS Biology

rhodge@plos.org

PLOS
